# Mind Reader:
# Reconstructing complex images from brain activities

**Sikun Lin**    **Thomas Sprague**    **Ambuj K Singh**
UC Santa Barbara
{sikun,tsprague,ambuj}@ucsb.edu

## Abstract

Understanding how the brain encodes external stimuli and how these stimuli can be decoded from the measured brain activities are long-standing and challenging questions in neuroscience. In this paper, we focus on reconstructing the complex image stimuli from fMRI (functional magnetic resonance imaging) signals. Unlike previous works that reconstruct images with single objects or simple shapes, our work aims to reconstruct image stimuli that are rich in semantics, closer to everyday scenes, and can reveal more perspectives. However, data scarcity of fMRI datasets is the main obstacle to applying state-of-the-art deep learning models to this problem. We find that incorporating an additional text modality is beneficial for the reconstruction problem compared to directly translating brain signals to images. Therefore, the modalities involved in our method are: (i) voxel-level fMRI signals, (ii) observed images that trigger the brain signals, and (iii) textual description of the images. To further address data scarcity, we leverage an aligned vision-language latent space pre-trained on massive datasets. Instead of training models from scratch to find a latent space shared by the three modalities, we encode fMRI signals into this pre-aligned latent space. Then, conditioned on embeddings in this space, we reconstruct images with a generative model. The reconstructed images from our pipeline balance both naturalness and fidelity: they are photo-realistic and capture the ground truth image contents well.

## 1 Introduction

In an effort to understand visual encoding and decoding processes, researchers in recent years have curated multiple datasets recording fMRI signals while the subjects are viewing natural images [3, 8, 34, 41]. In particular, the Natural Scenes Dataset (NSD [3]) was built to meet the needs of data-hungry deep learning models, sampling at an unprecedented scale compared to all prior works while having the highest resolution and signal-to-noise ratio (SNR). In addition, all the images used in NSD are sampled from MS-COCO [22], which has far richer contextual information and more detailed annotations compared to datasets that are commonly used in other fMRI studies (e.g., Celeb A face dataset [23], ImageNet [10], self-curated symbols, grayscale datasets). This dataset, therefore, offers the opportunity to explore the decoding of complex images that are closer to real-life scenes.

Human visual decoding can be categorized into stimuli category classification [1], stimuli identification [38], and reconstruction. We focus on stimuli reconstruction in this study. Different from previous efforts in reconstructing images from fMRI [6, 11, 12, 24, 32, 34, 35], we approach the problem with one more modality, that of text. The benefits of adding the text modality are threefold: first, the brain is naturally multimodal. Research [7, 13, 25] indicates that the brain is not only capable of learning multisensory representations, but a larger portion of the cortex is engaged in multisensory processing: for example, both visual and tactile recognition of objects activate the same part of the object-responsive cortex [26]. Visual-linguistic pathways along the border of the occipital lobe [27] also bring a more intertwined view of the brain's representation of these two modalities. Second, multimodal deep models tend to explain the brain better (having higher representation correlations)

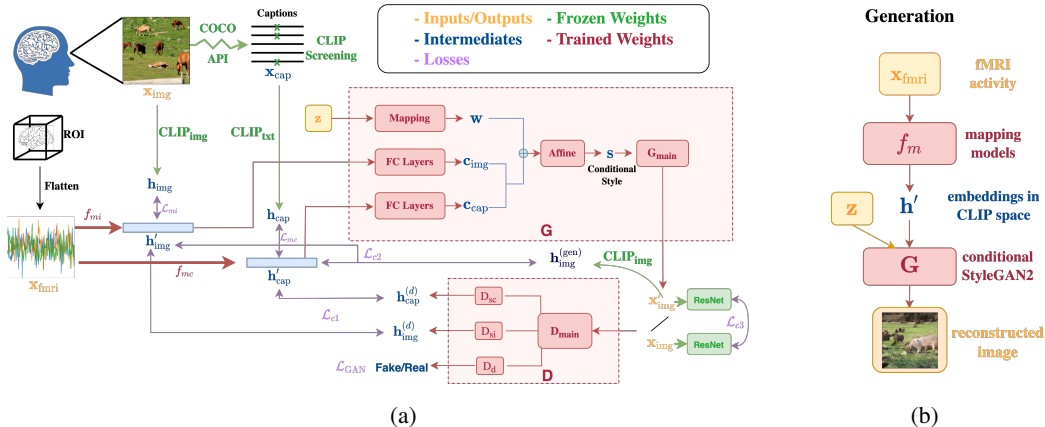

Figure 1: The pipeline for reconstructing seen images from fMRI signals. 1a details different components, from collected data to the reconstructed image. The pipeline is trained in two stages: during the first stage, mapping models $f_{mi}$, $f_{mc}$ are trained to encode fMRI activities into the CLIP embedding space. In the second stage, conditional generator $\mathbf{G}$ and contrastive discriminator $\mathbf{D}$ are finetuned while both $f_{mi}$, $f_{mc}$ are kept frozen. 1b shows the image generation process once models are trained.

than the visual-only models, even when compared with activities in the visual cortex [9]. Lastly, our goal is to reconstruct complex images that have multiple objects in different categories with intricate interactions: it is natural to incorporate contextual information as an additional modality.

Instead of training a model to map all three modalities (fMRI, image, text) to a unified latent space, we propose to map fMRI to a well-aligned space shared by image and text, and use conditional generative models to reconstruct seen images from representations in that space. This design addresses the data scarcity issue of brain datasets by separating fMRI from the other two modalities. In this way, a large amount of data is readily available to learn the shared visual-language representation and to train a generative model conditioned on this representation. Furthermore, pre-trained models can be utilized to make the whole reconstruction pipeline more efficient and flexible.

Our contributions are as follows: (1) to the best of our knowledge, this is the first work on reconstructing complex images from human brain signals. It provides an opportunity to study the brain's visual decoding in a more natural setting than using object-centered images. Compared to previous works, it also decodes signals from more voxels and regions, including those outside the visual cortex, that are responsive to the experiment. This inclusion allows us to study the behavior and functionality of more brain areas. (2) We address the data scarcity issue by incorporating additional text modality and leveraging pre-trained latent space and models. For the reconstruction, we focus on semantic representations of the images while taking low-level visual features into account. (3) Our results show we can decode complex images from fMRI signals relatively faithfully. We also perform microstimulation on different brain regions to study their properties and showcase the potential usages of the pipeline.

## 2 Method

Recent developments in contrastive models allow more accurate embeddings of images and their semantic meanings in the same latent space. This performance is realized using massive datasets: models such as CLIP [28] and ALIGN [17] utilize thousands of millions of image-text pairs for representation alignment. In comparison, brain imaging datasets that record pairs of images and fMRI range from 1.2k to 73k samples, making it difficult to learn brain encoding and decoding models from scratch. However, we can utilize aligned embeddings obtained from pre-trained contrastive models as the intermediary and generate images conditioned on these embeddings. In our pipeline, as shown in fig. 1, we first map fMRI signals to CLIP embeddings of the observed image and its captions, then pass these embeddings to a conditional generative model for image reconstruction.

### 2.1 Caption screening

Each image $\mathbf{x}_{\text{img}}$ in the COCO dataset has five captions $\{\mathbf{x}_{\text{cap}_1}, \cdots, \mathbf{x}_{\text{cap}_5}\}$ collected through Amazon's Mechanical Turk (AMT), and in nature, these captions vary in their descriptive ability. Fig. 2

| | **Captions** | **CLIP probabilities** |
|---|---|---|
| | (1) A group of people sitting and standing on top of a sandy beach. | 0.0376 |
| | (2) A surfboard rests on the beach while people play in the waves. | 0.519 ✔ |
| | (3) A surfboard on the sand and people on the beach behind. | 0.4302 ✔ |
| | (4) A few people are hanging out and appreciating their time. | 0.001286 |
| | (5) A group of people are sitting on the beach shore. | 0.0122 |

Figure 2: Image caption screening through CLIP encoders. For this sample, threshold is put at half of the largest probability: $0.5 \times 0.519$. Therefore, captions (2) and (3) of the image are kept.

shows a sample image with its five captions, and we can tell captions (2) and (3) are more objective and informative than caption (4) when it comes to describing the *content* of that image, thus are more helpful to serve as the image generation condition. We utilize pre-trained CLIP encoders to screen the high-quality captions since representations in the CLIP space are trained to be image-text aligned. A caption with an embedding more aligned to the image embedding is more descriptive than a less aligned one; it is also less general and more specific to this particular image because of the contrastive loss in CLIP. For the screening, we pass each image together with its five captions to the CLIP model, which outputs corresponding probabilities that the captions and image are proper pairs. We keep captions with probabilities larger than half of the highest probability. After screening out less informative captions, we have one to three high-quality captions per image.

## 2.2 Mapping fMRI signals to CLIP space

Each fMRI signal that reflects a specific image is a 3D data volume, and the value on position $(i, j, k)$ is the relative brain activation on this voxel triggered by the image. We apply an ROI (region of interest) mask on this 3D volume to extract signals of cortical voxels that are task-related and have good SNRs. The signal is then flattened into a 1D vector and voxel-wise standardized within each scan session. The end results $\mathbf{x}_{\text{fmri}}$ are used by our image reconstruction pipeline. We choose to use the ROI with the widest region coverage, and the length $N$ of $\mathbf{x}_{\text{fmri}}$ ranges from 12682 to 17907 for different brains in the NSD dataset.

Our goal is to train two mapping models, $f_{mi}$ and $f_{mc}$ in fig. 1 (collectively denoted as $f_m$), that encodes $\mathbf{x}_{\text{fmri}} \in \mathbb{R}^N$ to $\mathbf{h}_{\text{img}} = C_{\text{img}}(\mathbf{x}_{\text{img}}) \in \mathbb{R}^{512}$ and $\mathbf{h}_{\text{cap}} = C_{\text{txt}}(\mathbf{x}_{\text{cap}}) \in \mathbb{R}^{512}$ respectively. Here $C_{\text{img}}, C_{\text{txt}}$ are CLIP image and text encoders, and $\mathbf{x}_{\text{cap}}$ is one of the image captions chosen randomly from the vetted caption pool. We construct both $f_m$ as a CNN with one Conv1D layer followed by four residual blocks and three linear layers. The training objective is a combination of MSE loss, cosine similarity loss, and contrastive loss on cosine similarity. We use the infoNCE definition [39] of contrastive loss, for the $i^{\text{th}}$ sample in a batch of size $B$:

$$\text{Contra}(a^{(i)}, b^{(i)}) = -\mathbb{E}_i \left[ \log \frac{\exp(\cos(a^{(i)}, b^{(i)})/\tau)}{\sum_{j=1}^{B} \exp(\cos(a^{(i)}, b^{(j)})/\tau)} \right] \tag{1}$$

For the mapping model $f_{mi}$ that encodes fMRI to image embeddings, we have $\mathbf{h}_{\text{img}}^{(i)\prime} = f_{mi}(\mathbf{x}_{\text{fmri}}^{(i)})$. The training objective is:

$$\mathcal{L}_{mi} = \mathbb{E}_i \left[ \alpha_1 ||\mathbf{h}_{\text{img}}^{(i)\prime} - \mathbf{h}_{\text{img}}^{(i)}||_2^2 + \alpha_2(1 - \cos(\mathbf{h}_{\text{img}}^{(i)\prime}, \mathbf{h}_{\text{img}}^{(i)})) \right] + \alpha_3 \text{Contra}(\mathbf{h}_{\text{img}}^{(i)\prime}, \mathbf{h}_{\text{img}}^{(i)}), \tag{2}$$

where $\tau, \alpha_1, \alpha_2, \alpha_3$ are non-negative hyperparameters selected through sweeps. The loss $\mathcal{L}_{mc}$ for caption embedding mapping model $f_{mc}$ is defined similarly. Although CLIP embeddings are trained to be aligned, there are still systematic differences between image and text embeddings, with embeddings under each modality showing outlier values at a few fixed positions. In addition, we also notice the generated images emphasize either image content (object proximity, shape, etc.) or semantic features depending on which condition we use. Therefore, including both embeddings as the conditions for a generator can cover both ends, and that is why we train two mapping models for the two modalities. Since the outlier indices are fixed for each modality across images, clipping the value should not affect image-specific information. Therefore, before normalizing the ground truth embeddings into unit vectors, we set $\mathbf{h} = \text{clamp}(\mathbf{h}, -1.5, 1.5)$. This can greatly improve the mapping performance during training.

## 2.3 Image reconstruction with CLIP embedding conditioning

The mapping models output fMRI-mapped CLIP embeddings $\mathbf{h}'_{\text{img}}$ and $\mathbf{h}'_{\text{cap}}$ that serve as conditions for the generative model. We aim to generate images that have both naturalness (being photo-

realistic) and high fidelity (can faithfully reflect objects and relationships in the observed image). Our generation model is built upon Lafite [42], a text-to-image generation model: it adapts unconditional StyleGAN2 [20, 19] to conditional image generation contexted on CLIP text embeddings.

In our generator $\mathbf{G}$, both conditions $\mathbf{h}'_{\text{img}}$ and $\mathbf{h}'_{\text{cap}}$ are injected into the StyleSpace: each of them goes through two fully connected (FC) layers and is transformed into condition codes $\mathbf{c}_{\text{img}}$ and $\mathbf{c}_{\text{cap}}$. These condition codes are max-pooled and then concatenated with the intermediate latent code $\mathbf{w} \in \mathcal{W}$, which is obtained from passing the noise vector $\mathbf{z} \in \mathcal{Z}$ through a mapping network (see fig. 1). Using a mapping network to transform $\mathbf{z}$ into an intermediate latent space $\mathcal{W}$ is the key of StyleGAN as $\mathcal{W}$ is shown to be much less entangled than $\mathcal{Z}$ [36]. The conditioned style $\mathbf{s}$ is then passed to different layers of $\mathbf{G}$ as in StyleGAN2, generating image $\mathbf{x}_{\text{img}}'$:

$$\mathbf{s} = \mathbf{w} \| \max(\mathbf{c}_{\text{img}}, \mathbf{c}_{\text{cap}}), \quad \mathbf{x}_{\text{img}}' = \mathbf{G}(\mathbf{s}). \tag{3}$$

We align the semantics of generated $\mathbf{x}_{\text{img}}'$ and condition vectors by passing $\mathbf{x}_{\text{img}}'$ through pre-trained CLIP encoders and apply contrastive loss (eq. (5) $\mathcal{L}_{c2}$) between them. For further alignment of the lower-level visual features, such as prominent edges, corners and shapes, we also pass the image through resnet50 and align the position-wise averaged representation obtained from Layer2 (eq. (5)$\mathcal{L}_{c3}$).

The discriminator $\mathbf{D}$ has three heads that share a common backbone: the first head $\mathbf{D}_d$ classifies images to be real/fake, the second and the third semantic projection heads $\mathbf{D}_{si}$, $\mathbf{D}_{sc}$ map $\mathbf{x}_{\text{img}}'$ to $\mathbf{h}'_{\text{img}}$ and $\mathbf{h}'_{\text{cap}}$. The latter two ensure the generated images are faithful to the conditions. It is also shown that contrastive discriminators are useful for preventing discriminator overfitting and improving the final model performance [18, 16]. Applying contrastive loss (eq. (5) $\mathcal{L}_{c1}$) between the outputs from discriminator semantic projection heads and the condition vectors fed to $\mathbf{G}$ can therefore help stabilize the training. To summarize the objective function, the standard GAN loss is used to ensure the naturalness of generated $\mathbf{x}_{\text{img}}'$:

$$\begin{aligned}
\mathcal{L}_{\text{GAN}_{\mathbf{G}}} &= -\mathbb{E}_i \left[ \log \sigma(\mathbf{D}_d(\mathbf{x}_{\text{img}}^{(i)'})) \right], \\
\mathcal{L}_{\text{GAN}_{\mathbf{D}}} &= -\mathbb{E}_i \left[ \log \sigma(\mathbf{D}_d(\mathbf{x}_{\text{img}}^{(i)})) - \log(1 - \sigma(\mathbf{D}_d(\mathbf{x}_{\text{img}}^{(i)'}))) \right],
\end{aligned} \tag{4}$$

where $\sigma$ denotes the Sigmoid function. Meanwhile, contrastive losses are used to align the semantics of generated images and the fMRI-mapped condition vectors that supposedly residing in the CLIP space:

$$\begin{aligned}
\mathcal{L}_{c1} &= \text{Contra}(\mathbf{D}_{sc}(\mathbf{x}_{\text{img}}^{(i)'}), \mathbf{h}_{\text{cap}}^{(i)'}) + \text{Contra}(\mathbf{D}_{si}(\mathbf{x}_{\text{img}}^{(i)'}), \mathbf{h}_{\text{img}}^{(i)'}), \\
\mathcal{L}_{c2} &= \text{Contra}(\text{C}_{\text{img}}(\mathbf{x}_{\text{img}}^{(i)'}), \mathbf{h}_{\text{cap}}^{(i)'}) + \text{Contra}(\text{C}_{\text{img}}(\mathbf{x}_{\text{img}}^{(i)'}), \mathbf{h}_{\text{img}}^{(i)'}), \\
\mathcal{L}_{c3} &= \text{Contra}(\text{ResNet}(\mathbf{x}_{\text{img}}^{(i)'}), \text{ResNet}(\mathbf{x}_{\text{img}}^{(i)}))
\end{aligned} \tag{5}$$

The overall training objectives are: $\mathcal{L}_{\mathbf{G}} = \mathcal{L}_{\text{GAN}_{\mathbf{G}}} + \lambda_1 \mathcal{L}_{c1} + \lambda_2 \mathcal{L}_{c2} + \lambda_3 \mathcal{L}_{c3}, \mathcal{L}_{\mathbf{D}} = \mathcal{L}_{\text{GAN}_{\mathbf{D}}} + \lambda_1 \mathcal{L}_{c1}$, where $\lambda_1, \lambda_2, \lambda_3$, are non-negative hyperparameters.

The whole generation pipeline, consisting of mapping models and GAN, is trained in two stages. First, mapping models $f_{mi}$ and $f_{mc}$ are trained on fMRI-CLIP embedding pairs. Next, starting from the trained mapping model weights and Lafite language-free model weights, we modify the losses and model structure and finetune the conditional generator. For the additional condition vector projection layers in $\mathbf{G}$ and semantic head in $\mathbf{D}$, we duplicate the weights in the existing parallel layers to make the model converge faster. Note that Lafite is pre-trained on the Google Conceptual Captions 3M dataset [33] then finetuned on the MS-COCO dataset, both of which are much larger than NSD. Finetuning from it allows us to exploit the natural relationships between semantics and images with sparse fMRI data. We can still utilize a two-stage training to compensate for data scarcity even if no pre-trained conditional GAN like Lafite is available, for example, when using a different generator architecture. Only this time, we should firstly train the conditional GAN on a large image dataset with noise perturbed $\mathbf{h}_{\text{img}}$ and $\mathbf{h}_{\text{cap}}$ as the pseudo input condition vectors.

## 3 Results

### 3.1 Data and experimental setup

The NSD data is collected from eight subjects. We focus on reconstructing observed scenes from a single subject's brain signals. The reasons are twofold: first, it is more accurate to utilize individual

brain coordinates instead of mapping voxels into a shared space, which can result in information loss during the process. More importantly, brain encoding and perception are different among individuals. This project aims to get the best reconstruction for a single individual, thus training models on one subject's data. Nevertheless, the commonality of this encoding process among the population is an exciting topic for future explorations.

We use subject one from NSD: the available data contains 27750 fMRI-image sample pairs on 9841 images. Each image repeats up to three times during the same or different scan sessions. Note that brain responses to the same image can differ drastically during the repeats (fig. 7). The dataset is split image-wise: 23715 samples corresponding to 8364 images are used as the train set, and 4035 samples corresponding to the remaining 1477 images are used as the validation set. Therefore, our pipeline never sees the image it will be tested on during the training. We use 1pt8mm-resolution scans and only consider fMRI signals from voxels in the nsdgeneral ROI provided by the dataset. This ROI covers voxels responsive to the NSD experiment (voxels with high SNR) in the posterior aspect of the cortex, and contains 15724 voxels for subject one ($\mathbf{x}_{\text{fmri}} \in \mathbb{R}^{15724}$). Images are all scaled to $256 \times 256$. Additional experiment settings , including hyperparameters of two training phases, are provided in appendices A.1 and A.2. Our experiments are conducted on one Tesla V100 GPU and one Tesla T4 GPU. The code is publicly available.[1]

### 3.2 Mapping models from fMRI to CLIP embeddings

**Evaluation criteria**   In the first training stage, mapping models $f_{mi}$ and $f_{mc}$ are trained to encode fMRI signals to CLIP embeddings. We use two criteria to evaluate the mapper performance to decide which one to use in the next stage. The first criterion is FID (Fréchet Inception Distance) [14] between generated image and ground truth using the trained mapper and a pre-trained generator. Given a Lafite model pre-trained on MS-COCO (language-free setting), we can replace its conditional vector with the outputs of our mapping models to generate images conditionally. These FIDs can indicate the starting points of the finetuning processes: the lower the FID, the better the candidate model. Secondly, we use the success rate of image "retrieval" in a batch of size 300. For the $i^{th}$ sample in the batch, if the cosine similarity between $\mathbf{h}^{(i)'}$ and $\mathbf{h}^{(i)}$ is larger compared to between $\mathbf{h}^{(i)'}$ and $\mathbf{h}^{(j)}, j \neq i$, then it counts as one successful forward retrieval. For backward retrieval, we count the number of correct matches of $\mathbf{h}^{(i)}$ to all $\mathbf{h}^{(j)'}$.

**Configuration comparisons**   We tested different configurations on the mapping models, including: (1) Whether to place the threshold at $\pm 1.5$ as mentioned in section 2.2; (2) When training $f_{mi}$, whether to perform image augmentations before passing images through the CLIP encoder; (3) When training $f_{mc}$, whether to use the CLIP text embedding of a fixed caption, a random valid caption, or use the average embedding of all valid captions; (4) Which loss function to use: MSE only, cos (cosine similarity) only, Contra only, MSE + cos, MSE + cos + Contra; (5) Whether auxiliary networks help. We tested adding an auxiliary discriminator with GAN loss, as well as adding auxiliary expander networks with VICReg loss [4].

We found: (1) Clamping ground truth embeddings significantly increase performance; (2) Using image augmentations increase $f_{mi}$ performance. This further indicates CLIP embeddings are more semantic related; (3) For $f_{mc}$, selecting a random caption from the valid caption pool each time is better than using a fixed one or using the average embedding of all valid captions; (4) Using MSE + cos as the loss gives the best base models, but then finetune these base models with MSE + cos + Contra can further lower the starting FID for pipeline finetuning, making the training in the next stage converge faster; (5) Adding auxiliary networks and objectives will not improve the performance. In general, although $\mathbf{h}_{\text{cap}}$ and $\mathbf{h}_{\text{img}}$ are already relatively well aligned, $f_{mc}$ can still map $\mathbf{x}_{\text{fmri}}$ closer to $\mathbf{h}_{\text{cap}}$ than $\mathbf{h}_{\text{img}}$, whereas $f_{mi}$ maps $\mathbf{x}_{\text{fmri}}$ to an embedding that is equally close to both, while being able to capture a few extreme values in $\mathbf{h}_{\text{img}}$ (see appendix A.3 for numerical details and mapped embedding visualizations). We think this difference reflects that it is easier to map fMRI signals to a more semantic representation (from the text space) than to a visual one.

To verify fMRI-mapped embeddings $\mathbf{h}'$ are semantically well aligned with ground truth CLIP embeddings, we examined the mismatches during the image retrieval. For four incorrect retrievals, fig. 3 shows which images' $\mathbf{h}^{(j)'}$ are closer to the ground truth images' $\mathbf{h}^{(i)}$ than $\mathbf{h}^{(i)'}$. Notably, these mismatches are semantically close to the ground truth images. This indicates that the mapping models

---

[1]https://github.com/sklin93/mind-reader

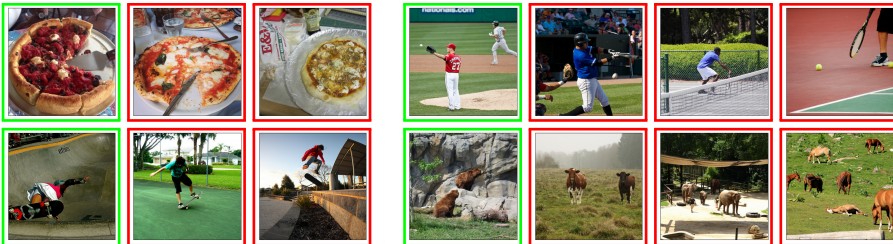

Figure 3: Mismatches are semantically close to the ground truth. Figure shows examples of incorrect matches $j$ (red frame) in a batch of 300 in the validation set. For each ground truth image $i$ (green frame), we pass it through CLIP encoder to get $\mathbf{h}^{(i)}$ and through $f_{mc}$ to get $\mathbf{h}^{(i)'}$. The shown incorrect ones are those images with $\mathbf{h}^{(j)'}, j \neq i$ that is closer to $\mathbf{h}^{(i)}$ than $\mathbf{h}^{(i)'}$.

Table 1: FID of the pipeline under different settings.

| FID↓ | | | $f_{mi}$ | $f_{mc}$ | $f_{mi}$ & $f_{mc}$ |
|---|---|---|---|---|---|
| from supervised | without $\mathcal{L}_{c3}$ | $f_m$ frozen | 37.75 | 41.51 | — |
| from LF | without $\mathcal{L}_{c3}$ | $f_m$ frozen | 30.83 | 33.78 | 50.59 |
| from LF | with $\mathcal{L}_{c3}$ | $f_m$ frozen | **29.74** | 33.35 | 49.47 |
| from LF | with $\mathcal{L}_{c3}$ | end to end | 45.02 | 48.54 | 50.96 |

can successfully map fMRI signals into a semantically disentangled space. Embeddings in this space are suitable for providing contexts to a conditional generative model. We also tested another mapping model $f_{mr}$ that maps fMRI signals to representations obtained from resnet50 $\mathrm{Layer}2$. Unlike the CLIP embedding space, the resnet vector encodes more lower-level visual features. We see a jump in the image retrieval rate when we combine the representations obtained from $f_{mi}$, $f_{mc}$ with $f_{mr}$ (table 4). However, the generative model is difficult to train when taking in two conditions from distinct embedding spaces. Therefore, we add the low-level vision constraint into the contrastive loss $\mathcal{L}_{c3}$ instead.

### 3.3 Conditional image generation

**Quantitative results**    In the second training stage, we finetune the conditional StyleGAN2.[2] There is no standard metric to measure image reconstruction quality from fMRI signals for complex images. Since previous works focused on reconstructing simpler images, the metrics typically involve pixel-wise MSE or correlation measures. However, when it comes to complex images, it seems more reasonable to use a perceptual metric, such as FID, which is based on Inception V3 [37] activations and is widely used in GAN. We also detail another metric, n-way identification accuracy, that reflects more of the fidelity and uniqueness of the generated images, in appendix A.4. We perform the ablation studies on the pipeline to answer the following questions: (1) Which mapping model trained in stage one leads to the best final performance? $f_{mc}$ or $f_{mi}$ or using both? (2) Which pre-trained GAN leads to the best final performance? For this, we compare using Lafite pre-trained on either the langue-free (LF) setting or the fully supervised setting. (3) Whether including the contrastive loss $\mathcal{L}_{c3}$ between lower-level visual features can further improve the performance of a semantic-based generative model? Finally, we tested (4) whether finetune the whole pipeline end-to-end or freezing the mapping models is better? The new mapping model losses are set to $\mathcal{L}'_m = \mathcal{L}_m + \lambda_4 \mathcal{L}_{\mathrm{GAN_G}}$ if trained end-to-end.

Results are reported in table 1. We observed the following: (1) in terms of FID, using $\mathbf{h}'_{\mathrm{img}}$ obtained from $f_{mi}$ as the generator condition is better than using the $\mathbf{h}'_{\mathrm{cap}}$ from $f_{mc}$ or using two conditional heads. On the other hand, $f_{mc}$ and the two-head setting achieve as good or even better performance as $f_{mi}$ does in terms of n-way identification accuracy. In addition, if training time or resource is the concern, using two heads and pre-trained LF-Lafite with only condition feeding interface changes and cloned weights in the new branches can already give reasonably good results. (2) Training the pipeline on LF-Lafite is much better than on the fully supervised Lafite. This result is expected for the generator conditioned on $\mathbf{h}'_{\mathrm{img}}$ since the supervised version is conditioned on CLIP text embeddings.

---

[2]Codes are adapted from https://github.com/NVlabs/stylegan2-ada-pytorch, https://github.com/drboog/Lafite

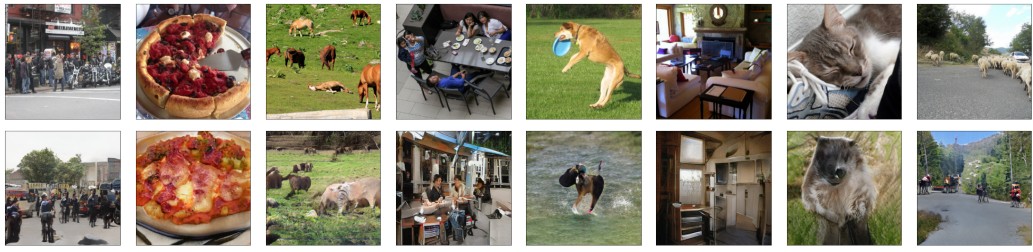

(a) Ground truth stimuli (top row) and generated images conditioned on fMRI (bottom row).

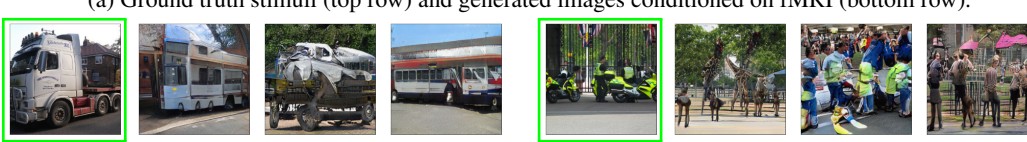

(b) Generated images from three different fMRI scans responding to the same stimulus (green frames).

Figure 4: Images generated by our pipeline given input fMRI signals.

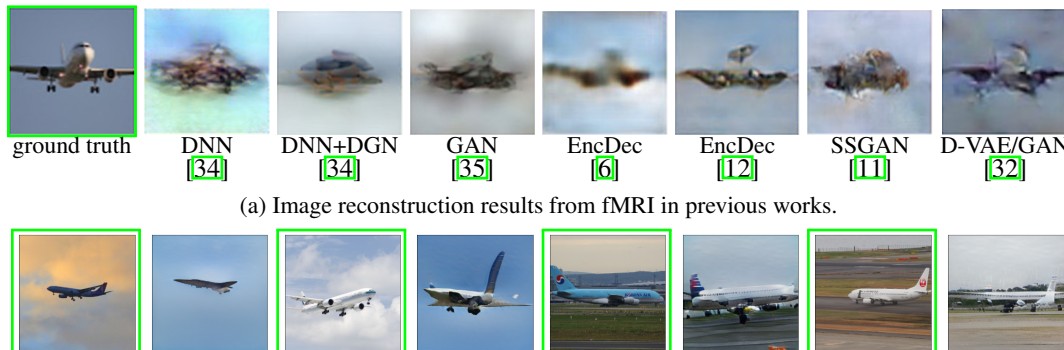

(a) Image reconstruction results from fMRI in previous works.

(b) Image reconstruction results from fMRI by our pipeline. Four ground truth images are green framed.

Figure 5: Comparisons between previous works and our pipeline. We are using the recent NSD dataset that involves more complex scenes. However, for comparison purposes, we choose four similar images from NSD, each containing a single object "plane", and show our reconstructions from fMRI signals in fig. 5b

However, the same discrepancy exists for the generator conditioned on $\mathbf{h}'_{\text{cap}}$. This may reflect the flexibility of pre-trained generators to adapt to the slight changes in the embedding space. It also shows the crucial impact of a pre-trained model on final performance when training data is limited. (3) The addition of low-level visual feature constraint $\mathcal{L}_{c3}$ is beneficial for the model performance, especially faithfulness. It also seems to have more effects on single-head models than the two-head one. (4) For the end-to-end pipeline training, we test performance with $\lambda_4 = [0.1, 1, 10]$, all of which give worse performance than keeping the mapping model weights frozen (reported values are from $\lambda_4 = 1$). In particular, we found that $\mathbf{h}'$ tends to collapse to having nonzero values at only a few positions if the mappers are finetuned together with GAN.

**Qualitative results** We show several generated images in fig. 4. Although the generator takes in both the noise vector $\mathbf{z}$ and fMRI-mapped embeddings, the results vary much more with the latter condition, while $\mathbf{z}$ only contributes to variations on some minor details. In general, the generated images capture both semantics and visual features relatively well, even on complex images containing interactions of multiple objects. Since each stimulus is repeated up to three times to the subject, we have multiple fMRI scans corresponding to the same image. The semantic differences in the generated images conditioned on these multiple scans could potentially reveal brain processing discrepancies of the same stimulus. For example, the three generations for the second image in fig. 4b emphasize respectively: (1) the overall scene and the fence, (2) people with green suits, and (3) overhead flags and the fence; these might reflect the variations in the subject's attention or interpretations of that image. Eyetracking data can be further examined to study attention's effect on generated images.

It is challenging to perform one-to-one comparisons with previous deep image reconstruction works since the images in the MS-COCO dataset have much higher complexities than artificial shapes, faces, or images containing a single centered object (like in ImageNet). We show results from a few best models for reconstructing images from fMRI in fig. 5a. There is also a recent survey [29] covering more models and results if readers are interested. As our dataset is different, we search for similar images in the NSD validation set and show our generations in fig. 5b. Compared to other methods, our pipeline can generate more photo-realistic images that reflect objects' shapes and backgrounds well. It also utilizes more voxel activities than previous works (15724 voxels versus a few hundred). More importantly, it is able to reconstruct the relationships of different components when the images are more complex. As natural scenes around us are rarely isolated objects and always information-laden, we think reconstructing images through semantic alignment and conditioning is more beneficial and realistic than focusing on lower-level visual features.

**Microstimulation** In neuroscience, microstimulation refers to the electrical current-driven excitation of neurons and is used to identify the functional significance of a population of neurons. Here, we "microstimulate" the input fMRI signals of voxels in different brain ROIs, aiming to identify the roles of individual regions. In the NSD dataset, there are four floc (functional localizer) experiments targeting regions responsible for faces, bodies, places, and words. A typical standardized fMRI signal has a value range around $[-4, 4]$. For the experiment, we locate the corresponding task-specific voxels based on ROI masks and increase the voxel activities to 10 while keeping the activities in unrelated voxels unchanged (see appendix A.5 for visual results). We observe the emergence of bodies or words when we increase the voxel activities in "bodies" or "words" ROIs. For voxels in "places" ROIs, elevating the signals will result in mesh-like patterns in the background, and this is true across different images. For "faces" ROIs, the generated images under elevated facial area signals seem to contain many small repeated patterns/perturbations. Interestingly, this appears to result from FFA (fusiform face area) signal changes since increasing only OFA (occipital face area) regions' activity does not result in similar patterns. Overall, increasing a specific task ROI's signal across fMRI samples results in CLIP embedding changes in similar positions. This means the disentangled space of CLIP embedding aligns well with how the human brain processes visual cues.

Apart from task-specific ROIs, we also changed brain region activities based on their roles in the visual processing hierarchy. We use the streams mask in the dataset to identify early visual cortex ROIs, intermediate ROIs, and higher-level ROIs. We then zero out voxels at each level. Our observations are: (1) when silencing the early visual cortex, objects and the whole scene are prone to be in dull colors, and objects tend to have sharp shapes. Meanwhile, the mapped embedding in the CLIP space will constantly have a lower value at almost all positions compared to mapped from unchanged signals . (2) Silencing the higher-level ROIs has the opposite effect: more colors, more shapes, and crowded scenes. This is reasonable since the lower-level visual regions will bring up all the details when they lack high-level control. This time, the embeddings in the CLIP space have values consistently higher than normal. Finally, (3) silencing the intermediate ROIs seems to have the least visual impact or CLIP embedding changes among the three. When compared to random controls (masking the same number of voxels at random locations), we found that masking random voxels has minimal effect on mapped CLIP vectors and reconstructed images, but masking early/higher-level visual cortex levels change the results drastically. We performed the above microstimulation experiments on our pipeline with existing ROIs; however, it is potentially helpful for testing new ROI definitions and hypotheses.

### 3.4 CLIP space as the intermediary

In this section, we show that multimodal embedding space, particularly the CLIP space, is beneficial for brain signal decoding. To this end, we trained a set of multi-label category classifiers to classify if a certain object category exists in the image based on the following inputs: (1) image-triggered fMRI $\mathbf{x}_{\text{fmri}} \in \mathbb{R}^{15724}$; (2) image CLIP embeddings $\mathbf{h}_{\text{img}} \in \mathbb{R}^{512}$; (3) CLIP embeddings mapped from image-triggered fMRI $\mathbf{h}'_{\text{img}} \in \mathbb{R}^{512}$; (4) image ResNet embeddings $\text{ResNet}(\mathbf{x}_{\text{img}}) \in \mathbb{R}^{2048}$ (obtained from Layer4, the final block before fully connected layers). All classifier models consist of 3 linear layers with ReLU activations in between, and finish with a Sigmoid activation. For fMRI signals, we use (2048, 512) as the hidden dimension; for CLIP embeddings (setting (2) and (3)), we use (384, 256) as hidden dimensions; and for the ResNet embedding, we use (512, 256) as hidden dimensions. The final output covers 171 classes, including 80 things categories (bounded objects, like "person",

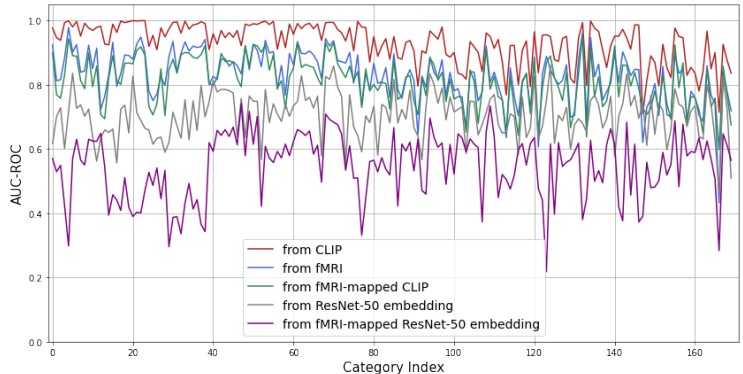

Figure 6: Category-wise AUC-ROC of multi-label classifiers that predicts from five different signal / embedding sources. The first 80 categories are "things categories" and the last 91 are "stuff categories" in COCO.

Table 2: Numerical AUC-ROC values of the classifiers presented in fig. 6.

| AUC-ROC | "things" categories | "stuff" categories | Overall | Performance w.r.t. fMRI (%) |
|---|---|---|---|---|
| CLIP | $0.9718 \pm 0.0266$ | $0.8973 \pm 0.0639$ | $0.9318 \pm 0.0624$ | 112.36 |
| fMRI | $0.8704 \pm 0.0557$ | $0.7937 \pm 0.0824$ | $0.8293 \pm 0.0807$ | 100.00 |
| fMRI-mapped CLIP | $0.8468 \pm 0.0604$ | $0.7817 \pm 0.0733$ | $0.8119 \pm 0.0748$ | 97.90 |
| ResNet-50 | $0.7061 \pm 0.0736$ | $0.7032 \pm 0.0719$ | $0.7044 \pm 0.0725$ | 84.94 |
| fMRI-mapped ResNet-50 | $0.5410 \pm 0.1106$ | $0.5520 \pm 0.0941$ | $0.5469 \pm 0.1020$ | 65.95 |

"car"), and 91 stuff categories (mostly unbounded objects, like "tree", "snow").[3] Binary cross-entropy loss is used for each class to predict its existence in the input image.

Fig 6 shows the category-wise AUC-ROC. The result demonstrates that CLIP embeddings contain the most object-level information about the image out of all the input sources. Following it, fMRI signals are also surprisingly very predictive, considering they carry a lot of noise. The performance discrepancy between settings (2) and (3) is minimal, meaning mapping fMRI signals into the CLIP space retains most of the fMRI signals' information: this provides strong support for the validity of our design. Lastly, ResNet embeddings perform poorly compared with other input sources. Therefore, even with a perfect mapping model, projecting fMRI signals into this space will lose information about the image since the expressiveness of the embedding is bounded by the lower performer. In addition, we note that both CLIP embeddings and fMRI have poorer performance on stuff categories than on things categories, whereas ResNet embeddings do not. This can indicate brain signals align better with the multimodal CLIP space than with single-modality ResNet space. Previous brain signal decoding work utilizing pre-trained generators all relied on image-only embedding spaces (ResNet-50 [24], VGG19 [35]), and we believe moving to a multimodal latent space is a crucial step towards better brain signal decodings.

## 4 Further Discussions

Prior to our current pipeline design, we experimented with a DALL-E-like structure [31] since we can view the image reconstruction problem as signal-to-signal translation. In particular, we applied VQVAE [40] on both fMRI and image to represent them as discrete latent codes and train a Transformer model to autoregressively generate text and image tokens from fMRI tokens. However, it was challenging to train the Transformer-based model to converge with limited fMRI-image data. Incorporating the caption as text tokens to serve as the bottleneck between fMRI and image tokens while utilizing pre-trained models on the text and image modality did not help either. We think this suggests the need to introduce a semantic medium to avoid direct translations between fMRI and image, as well as a solution to data scarcity.

We address both issues with the semantic space of CLIP embeddings. First, CLIP space is semantically informative and visually descriptive: for example, we can use image-text CLIP embedding alignment

---

[3]Please refer to https://github.com/nightrome/cocostuff/blob/master/labels.txt for the full category list.

probabilities to screen captions. Mapping fMRI signals to representations in this latent space will retain rich information about the image that needs to be reconstructed. Second, the pre-training of the generative model can be separated entirely from fMRI data, meaning it can utilize much larger datasets than the one we use. However, there is a trade-off between generating a semantically similar scene and faithfully reconstructing each pixel. Although trained with additional contrastive loss targeting low-level visual features, the generated images by our pipeline are still leaning towards the former. We consider this a reasonable choice since brains are more likely to perceive the image as a whole rather than identifying each pixel, especially with multiple objects in the scene. Nevertheless, this results in worse reconstructions for images with fine details but less semantic, such as single faces. The reconstruction of complex images with better aligned low-level visual features is worth further studies.

There are many more areas to explore. First, our study focuses on reconstructing a single subject's brain signals. Applying the model to different subjects and observing the differences when generating the same image would be interesting. Since the data contains behavioral measures like valence and arousal towards each image, one can test if the generated images reflect personalized attention and perceptions. Second, other latent spaces can be examined. Although CLIP is one of the best-aligned computation models for the brain, other multimodal models like TSM [2] seem to have a better alignment [9] with the visual cortex. In addition, other conditional generative models, such as diffusion models, can be explored. In particular, DALL-E 2 [30] generates images conditioned on CLIP embeddings through diffusion, and it also provides an alternative solution to the differences exhibited in the image text CLIP embeddings by learning a *Prior* model. Third, given the additional text modality, our pipeline opens up new opportunities to study visual imagery even without ground truth images. For example, one can either use mapping models trained on given fMRI-image pairs and pre-trained generators to reconstruct imagined scenes, or study the mapping between brain signals and the text embeddings of the mental images' descriptions. Lastly, we focus on the decoding (brain-to-image) process, but the encoding (image-to-brain) process of complex images is equally important and exciting (we provide initial results on encoding in appendix A.9; additional future directions are discussed in appendix A.10).

With current brain signal recording devices, the negative social impact of this work is minimal: portable devices like EEG have poor spatial resolutions, making them unlikely to provide enough image-related details; On the other hand, fMRI scanners are used under highly controlled settings with designed procedures, therefore unlikely to have subject-unapproved privacy violations. However, when new devices that can address these issues become readily available, regulations would be needed on collecting and inspecting user data, since they potentially reveal sensitive information that users are unwilling to share through neural decoding. With pre-trained components, the pipeline may also misinterpret brain signals or be hacked to generate from manipulated inputs (no matter how unlikely it is) and produce over-confident false reconstructions because of the training data distributions. Several tricks may alleviate this issue, for example, training an input discriminator and placing it before the entire pipeline to filter out suspicious inputs. Or, using a parallel pipeline targeting pixel-level reconstruction as a check: if the two systems agree with each other above a certain threshold/confidence, the reconstruction results are accepted, otherwise discarded. Future pipeline improvements should also focus on exploring high-performing models pre-trained on large (thus more generalizable) and unbiased datasets.

## 5   Conclusion

The paper proposes a pipeline to reconstruct complex images observed by subjects from their brain signals. With more objects and relationships presented in the image, we bring in an additional text modality to better capture the semantics. To achieve high performance with limited data, we utilize pre-trained semantic space that aligns visual and text modalities. We first encode fMRI signals to this visual-language latent space and use a generative model conditioned on the mapped embeddings to reconstruct the images. We also introduce additional contrastive loss to incorporate low-level visual features into this semantic-based pipeline. As a result, the reconstructed images by our method are both photo-realistic and, most of the time, can faithfully reflect the image content. This brain signal to image decoding pipeline opens new opportunities to study human brain functions through strategic input alterations and can even potentially be helpful for human-brain interfaces.

## Acknowledgments and Disclosure of Funding

This project was partially supported by National Science Foundation under IIS-1817046 and HDR DSC 1924205.

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
