# A  Appendix

## A.1  Data

**fMRI data**    fMRI activities differ when the same individual sees the same image at different times (fig. 7). Although we use *activities* and *signals* interchangeably throughout the paper, what we mean are fMRI *betas* in the NSD dataset. Betas are not direct measurements of BOLD (blood-oxygenation-level dependent) changes, but the inferred activities from BOLD signals through GLM (general linear models). The reason for using betas instead of direct measurements is that image stimuli are shown consecutively to the subjects without prolonged delay, and activities triggered by the previous image can interfere with the next one if there is no proper separation. Authors of NSD proved the effectiveness of their GLM approach with much improved SNR in the betas over raw measurements [3].

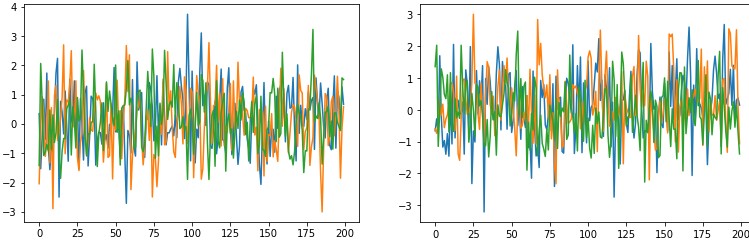

Figure 7: fMRI activities responding to two images, each repeating three times. The figure only shows the activities of the first 200 voxels for visualization purposes.

**Image augmentation during training**    Based on conclusions from StyleGAN2-ADA [19], we perform the following image augmentations before passing images into the CLIP encoder when training the fMRI-CLIP mapping model:

- perform random sized crop with a scale between 0.8 to 1.
- perform horizontal flip with probability $p = 0.5$.
- perform $ColorJitter(0.4, 0.4, 0.2, 0.1)$ with $p = 0.4$.
- perform grayscale with $p = 0.2$.
- perform Gaussian blur with $p = 0.5$ and kernel size 23.
- perform random masking with 0.3 masking ratio.

We test mapping models trained with and without the above augmentations, and found augmentations can improve fMRI to CLIP image embedding mapping performance (details are in table 3).

**CLIP embeddings and thresholding**    See fig. 8 for the visualizations of CLIP embeddings that show image and text embedding differences, effects of thresholding, image augmentation, and random caption selection.

## A.2  Experiment hyperparameters

The following hyperparameters are used in our experiments:

- $\tau = 0.5$ in eq. (1) for all the contrastive losses.
- for fMRI-CLIP mappers $f_{mi}, f_{mc}$ (losses are in eq. (2)), the models are first trained with $\alpha_1 = 0.4, \alpha_2 = 0.6, \alpha_3 = 0$, then finetuned with $\alpha_1 = 0.2, \alpha_2 = 0.3, \alpha_3 = 0.5$.
- mappers are trained with batch size 32 (on a single GPU) when not including contrastive loss, and batch size 128 when including contrastive loss or using VICReg loss. Learning rate is 0.0004.
- $\lambda_1 = 5, \lambda_2 = 10, \lambda_3 = 10$ for the losses of conditional StyleGAN2.
- conditional StyleGAN2 is trained with batch size $16 \times$ number of GPUs (in our case $B = 32$ since we used two GPUs). Learning rate is 0.0025.

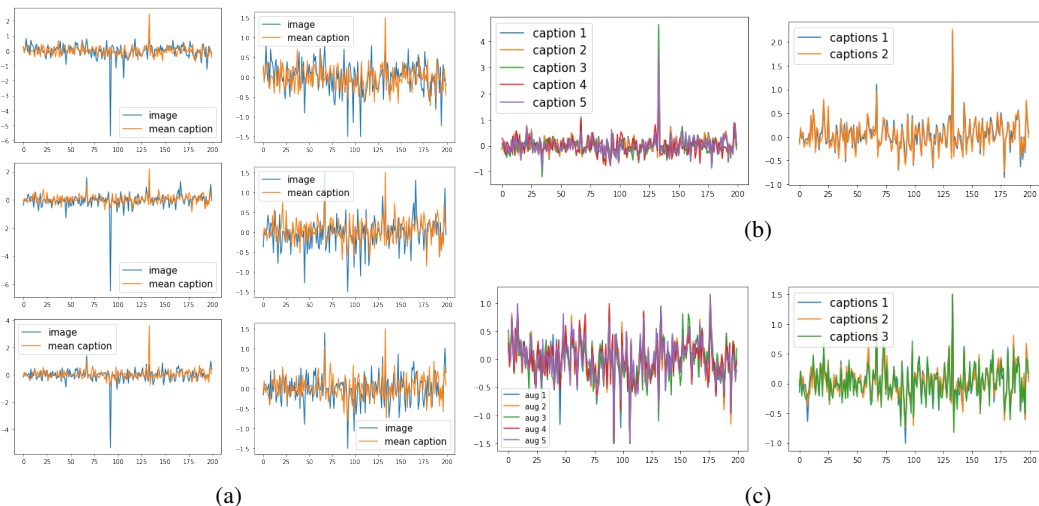

(a)                                                                    (b)

(c)

Figure 8: CLIP vector visualizations and thresholding. 8a: before (left column) v.s. after (right column) thresholding at $\pm 1.5$ to remove outliers. There are **systematic differences between CLIP image embeddings** and text embeddings; the outliers typically occur at the same positions for each modality. 8b: the caption screening process can make the kept caption embeddings more aligned. (b)1 and (b)2 are from the same sample, only difference is the screening process. 8c: (thresholded) embeddings of the same image with different augmentations; embeddings of same image's different screened captions. All embeddings are shown the first 200 values for visualization purposes.

## A.3   Results for the fMRI-CLIP mapping models $f_m$

Mapping models $f_{mi}$ and $f_{mc}$ are trained under different settings detailed in section 3.2, here we list the numerical results of the summarized findings in table 3. Simply put, forward retrieval checks the correct match of "*which ground truth CLIP embedding is the closest to the fMRI-mapped one?*" while the backward retrieval checks "*which fMRI-mapped embedding is the closest to the ground truth CLIP one?*". When multiple losses are involved, we use hyperparameter settings as in A.2.

Fig. 9 visualizes the mapping results of the best setting (models trained with threshold, image augmentation, use a random valid caption each time, pre-trained with MSE+cos loss then finetuned with MSE+cos+Contra loss).

Combining the mapped embeddings from multiple mappers boosts the retrieval performance, especially the backward one (as shown in table 4). To use multiple mapping models, we first calculate a $B \times B$ batch similarity matrix between the mapped embeddings for each model. Then we combine the similarity matrices with a weighted sum (weights are obtained through grid search) and perform image retrievals based on this combined similarity matrix. The mapping model $f_{mr}$ that encodes fMRI to ResNet embeddings has a correct forward retrieval 6 and backward retrieval 50. But when its similarity matrix is combined with mapped-CLIP embedding similarity matrices, the performance is far above that of both ResNet and CLIP embeddings.

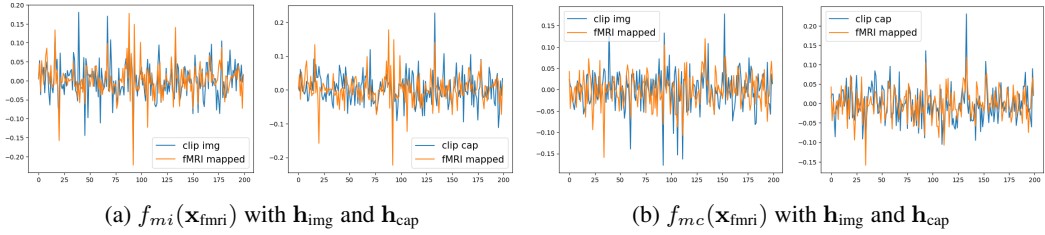

(a) $f_{mi}(\mathbf{x}_{\text{fmri}})$ with $\mathbf{h}_{\text{img}}$ and $\mathbf{h}_{\text{cap}}$          (b) $f_{mc}(\mathbf{x}_{\text{fmri}})$ with $\mathbf{h}_{\text{img}}$ and $\mathbf{h}_{\text{cap}}$

Figure 9: Embeddings mapped from fMRI signals overlay on ground truth CLIP embeddings. fig. 9a shows the results of image embedding mapping model $f_{mi}$, and fig. 9b shows the results of caption embedding mapping model $f_{mc}$. For visualization purposes, the figures only show the first 200 values of the length-512 vectors.

Table 3: Starting FID without generator finetuning (pre-trained LF-Lafite is used here) and correct retrievals in a batch of size 300 using embeddings obtained from $f_{mi}$ and $f_{mc}$. In the top table, models are trained with MSE+cos loss. In the bottom table, defaults are: with threshold, with image augmentation, using random caption. For the two options with auxiliary modules, the model is finetuned from MSE + cos model since training from scratch gives much worse results. FID evaluations are omitted if the retrieval performance of a setting is strictly worse than its competitors.

|  |  | **threshold** | no threshold | **image aug** | no image aug | fixed caption | **random caption** | average caption embedding |
|---|---|---|---|---|---|---|---|---|
| $f_{mi}$ | FID↓ | 73.46 | — | 73.46 | — | n/a | n/a | n/a |
|  | Retrieval (forward)↑ | 21 | 13 | 21 | 19 | n/a | n/a | n/a |
|  | Retrieval (backward)↑ | 49 | 25 | 49 | 46 | n/a | n/a | n/a |
| $f_{mc}$ | FID↓ | 75.24 | — | n/a | n/a | — | 75.24 | 79.36 |
|  | Retrieval (forward)↑ | 14 | 11 | n/a | n/a | 13 | 14 | 15 |
|  | Retrieval (backward)↑ | **64** | 45 | n/a | n/a | 39 | **64** | 43 |

|  |  | MSE | cos | Contra | MSE + cos | MSE + cos + Contra (from scratch) | **MSE + cos + Contra (from MSE + cos)** | Auxiliary GAN | Auxiliary expander (VICReg) |
|---|---|---|---|---|---|---|---|---|---|
| $f_{mi}$ | FID↓ | — | — | — | 73.46 | — | **68.14** | — | — |
|  | Retrieval (forward)↑ | 5 | 12 | 25 | 21 | 27 | **29** | 25 | 19 |
|  | Retrieval (backward)↑ | 16 | 34 | 50 | 49 | 50 | **51** | 42 | 35 |
| $f_{mc}$ | FID↓ | — | — | — | 75.24 | — | **53.68** | — | — |
|  | Retrieval (forward)↑ | 4 | 10 | 27 | 14 | 30 | **33** | 24 | 9 |
|  | Retrieval (backward)↑ | 19 | 31 | 42 | **64** | 43 | 45 | 38 | 37 |

Table 4: Correct image retrievals in a batch of size 300 when combining different models.

| Multiple models | $f_{mi} + f_{mc}$ | $f_{mi} + f_{mc} + f_{mr}$ |
|---|---|---|
| Retrieval (forward) | 32 | 24 |
| Retrieval (backward) | 73 | 147 |

## A.4 Additional quantitative results (generator)

In addition to using FID as a metric, we also perform 2-way identification for images reconstructed by models under different settings, and n-way identification of generated images with $n = 2, 5, 10, 50$ under the best setting (finetuned from LF, with $\mathcal{L}_{c3}$, with mapping models $f_m$ frozen). For n-way identification, we reconstruct an image from the fMRI signal for each sample in the validation set. For each generated image, we compare it with a set of $n$ randomly selected images, including the ground truth one. Then based on the cosine similarity of their Inception V3 embeddings (before FC layers, the length-2048 vector), we identify which image the generated one corresponds to. This process is repeated ten times because of the randomness of the n-sample selection. Results are reported in tables 5 and 6. The n-way identification accuracy of the two-head setting ($f_{mi}$ & $f_{mc}$) is slightly better most of the time (table 6), followed by the caption-vector-conditioned setting, followed by the image-vector-conditioned setting. Note that when performing n-way identification, previous image reconstruction works are typically tested on a validation set that contains 50 images of 50 different

Table 5: 2-way identifications accuracy of the pipeline under different settings.

| accuracy (%) |  |  | $f_{mi}$ | $f_{mc}$ | $f_{mi}$ & $f_{mc}$ |
|---|---|---|---|---|---|
| from supervised | without $\mathcal{L}_{c3}$ | $f_m$ frozen | $72.6 \pm 6.14$ | $68.6 \pm 5.22$ | — |
| from LF | without $\mathcal{L}_{c3}$ | $f_m$ frozen | $73.0 \pm 4.40$ | $73.2 \pm 4.49$ | $76.2 \pm 5.89$ |
| **from LF** | **with $\mathcal{L}_{c3}$** | **$f_m$ frozen** | $76.8 \pm 4.16$ | **$78.2 \pm 5.47$** | $78.0 \pm 4.47$ |
| from LF | with $\mathcal{L}_{c3}$ | end to end | $51.4 \pm 5.59$ | $50.8 \pm 5.43$ | $50.2 \pm 5.31$ |

Table 6: n-way identification accuracy (%) with $n = 2, 5, 10, 50$.

| $n$ | 2 | 5 | 10 | 50 |
|---|---|---|---|---|
| $f_{mi}$ | $76.8 \pm 4.16$ | $55.2 \pm 3.23$ | $41.9 \pm 6.09$ | $24.9 \pm 3.98$ |
| $f_{mc}$ | $\mathbf{78.2 \pm 5.47}$ | $56.4 \pm 3.32$ | $42.2 \pm 4.33$ | $25.6 \pm 4.05$ |
| $f_{mi}$ & $f_{mc}$ | $78.0 \pm 4.47$ | $\mathbf{57.3 \pm 3.63}$ | $\mathbf{44.0 \pm 6.05}$ | $\mathbf{25.8 \pm 3.82}$ |

categories [15]. However, there are multiple objects involved in each image in the complex images we aim to reconstruct; it is not straightforward to separate them into different categories and pick one from each. Therefore, we leave the validation set as is (1477 image-fMRI pairs in total), and there will be overlapping categories in it; for example, several images contain scenes of animals in a natural environment.

## A.5 Visual results from microstimulation experiments

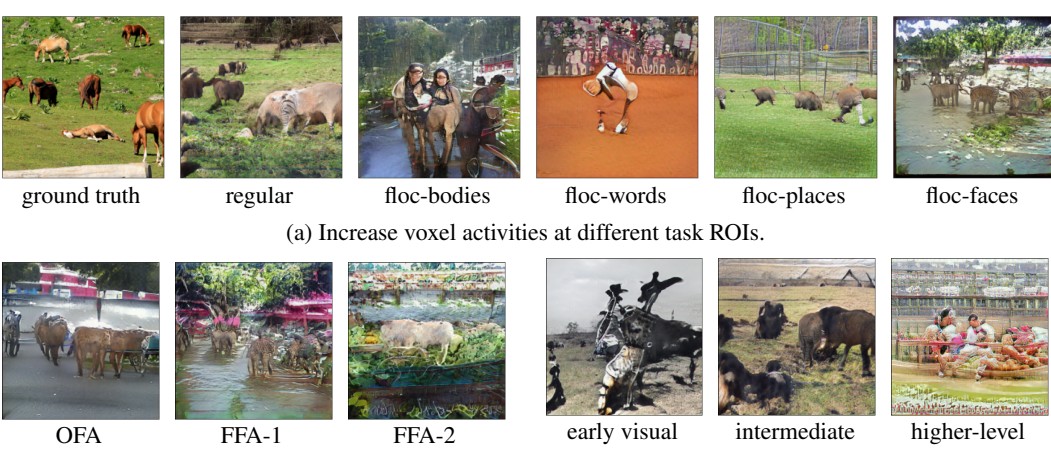

(a) Increase voxel activities at different task ROIs.

(b) Increase voxel activities at different face areas. (c) Set voxel activities to 0 at different processing levels

Figure 10: Images generated in microstimulation experiments. In 10a 10b, voxel activities at multiple task ROIs are increased before passed into the pipeline. In 10c, voxel activities at various visual processing stages are silenced.

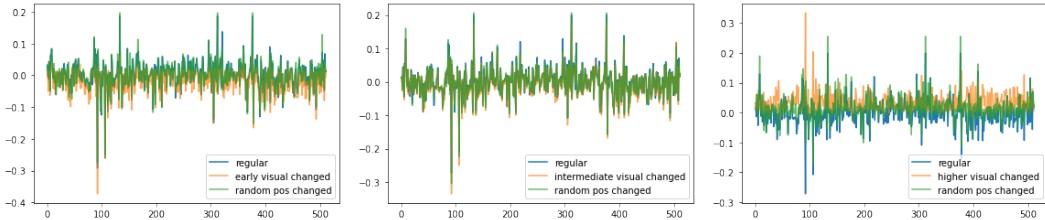

Figure 11: fMRI-mapped embeddings in the CLIP space ($\mathbf{h}'$). Each figure contains (i) an embedding mapped from a regular fMRI signal, (ii) an embedding mapped from the fMRI signals with voxel activities in earlier-visual ROIs (left)/ intermediate ROIs (middle) / higher-level ROIs (right) set to zero, (iii) an embedding mapped from the fMRI signal with voxels at random positions (same number of voxels as (ii)) set to zero. Setting activities of the earlier-visual cortex to zero lowers overall embedding vector values, while setting activities of higher-level ROIs has the opposite effect. We can also perform the reverse masking: only keep voxel activities at earlier-level visual/ intermediate / higher-level ROIs, then the effects are reversed.

Fig 10 shows generated images under different microstimulation experiments. Fig 11 shows the results regarding changes of mapped fMRI embeddings in the CLIP space when perturbing voxels in different visual cortex levels.

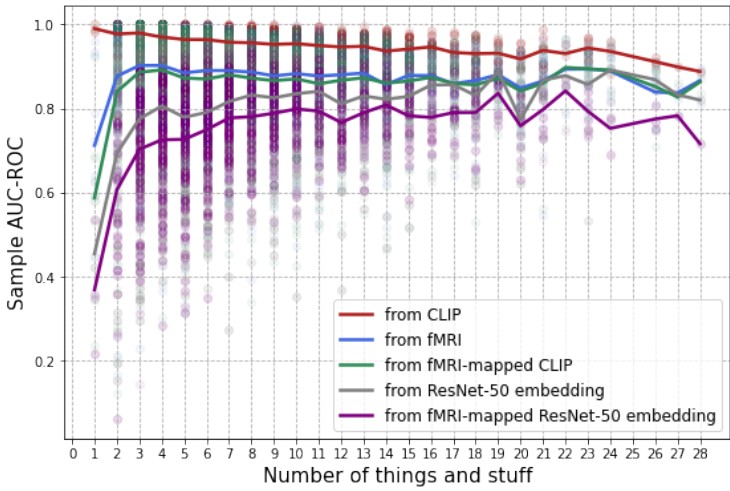

Figure 12: Sample-wise AUC-ROC of multi-label classifiers that predicts from five different signal/embedding sources as the number of samples in stimulus images increases.

## A.6 Additional result on using CLIP space as the intermediary

Apart from the alignment between brain signals and CLIP embeddings discussed in section 3.4, we also found that when the number of objects in the image increases, per-sample classification performance using CLIP, fMRI, and fMRI-mapped CLIP vector as inputs gradually decreases (the only difference is the single-object case). In contrast, ResNet inputs do not exhibit this property (fig. 12). We hypothesize that CLIP vectors can better mimic the cognitive overload when the scene becomes more crowded.

## A.7 Using pre-trained models

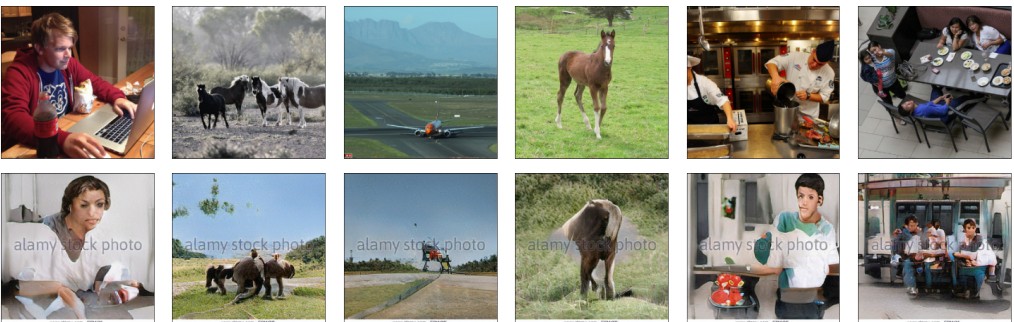

Figure 13: Image generated by Lafite pre-trained on the CC3M dataset without finetuning on COCO or NSD. Ground truth stimuli (top row) and generated images conditioned on fMRI (bottom row).

Our pipeline relies on two pre-trained components. The first and the most crucial one is the CLIP encoder that provides the latent space where we project fMRI signals. The second is a conditional GAN (Lafite) that generates images, which could be swapped for other generators. In what follows, we will discuss these two components separately.

**CLIP** One exciting aspect of CLIP is the size of its training dataset, which consists of 30 million Flicker images that should cover most of the natural image statistics. This coverage is also proved by subsequent works that generate images guided by CLIP embeddings through their abilities to perform generations in various styles. In addition, as we observed in 3.4, CLIP embeddings can retain around 98% of object-level information in fMRI with a very well-aligned performance across categories.

Albeit its incredible expressive power, CLIP does have a much lower dimensionality than the original signal: no matter how faithful, it is a compression. By the nature of compression, CLIP only

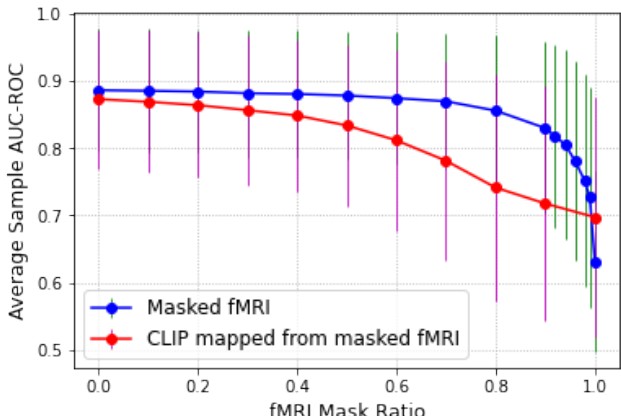

Figure 14: Multi-label classifier (defined in 3.4)'s average sample-wise AUC-ROC changes when masking input fMRI at different ratios. For a masked voxel, we set its value to 0.

retains the most crucial information and removes most of the redundancies in the original signal. Indeed, if we mask fMRI at different ratios, from 0 to 1, and perform the multi-label classification (the same task as in 3.4) using (1) masked fMRI or (2) CLIP mapped from masked-fMRI, we will notice a very drastic difference in the performance drop rate. As shown in fig. 14, prediction performance from fMRI only drops drastically after the masking ratio becomes larger than 0.9, indicating brain redundancies to represent the objects (refer [21] for further studies regarding brain signal redundancies). In contrast, if we map the masked fMRI into the CLIP space and use these embeddings for prediction, the performance drop is almost at a constant rate. This discrepancy makes the CLIP space more vulnerable to adversarial attacks than the fMRI space since a small change would cause the generated images to derail from the ground truth. In addition, CLIP embeddings also carry more biases than fMRI, as its mean AUC-ROC is much larger even with all-masked inputs. One should consider these traits of CLIP embeddings when applying this system and design defense mechanisms accordingly.

**Lafite** As for the generator, we utilize a conditional GAN pre-trained on the MS-COCO dataset (containing 328K images), from which NSD drew its experiment images. This naturally provides an alignment in the data distribution. Although MS-COCO images are about everyday objects, humans, and scenes, the data statistics could vary when we move to other settings. Therefore, future studies are needed to extend current generators to one trained on broader sources (e.g., DALL-E 2, mentioned in section 4, used 650M images sampled from CLIP and DALL-E training data). This should minimize the dataset biases, although one should not interpret results without considering the training/testing discrepancies.

To show that our concept works across different generators, but dataset biases indeed play an important role, we test our pipeline with a Lafite pre-trained on the Google Conceptual Captions 3M dataset (CC3M, consisting of 3.3 million images) as the generator *without any extra finetuning*. We used our trained $f_{mc}$ as the mapping function. The results are shown in fig. 13. All generated images have the watermark where CC3M sources its images. In addition, when trying to generate out-of-distribution images, the quality decreases in terms of photo-realism. Nonetheless, semantic alignments are still shown in these reconstructions. We also want to note that pre-trained models provide excellent bases for finetuning. For example, Lafite finetuned its COCO model on the CC3M model within three hours, compared to four days to reach the same performance if training from scratch. Therefore, if the pipeline is known to be used on certain types of images, a small-scale dataset and some light training should greatly help the model to fit into the desired data distribution.

## A.8 Additional examples

As mentioned in section 2.2, we found that the generated results conditioned on embeddings of different modalities tend to emphasize different aspects: more visual (colors, shape, etc.) if conditioned only on $\mathbf{h}'_{\text{img}}$, and more semantic if conditioned only on $\mathbf{h}'_{\text{txt}}$. This could reflect the slight difference between the latent space of the two modalities. We show the examples conditioned on either one of these two conditions, or both, in fig. 15.

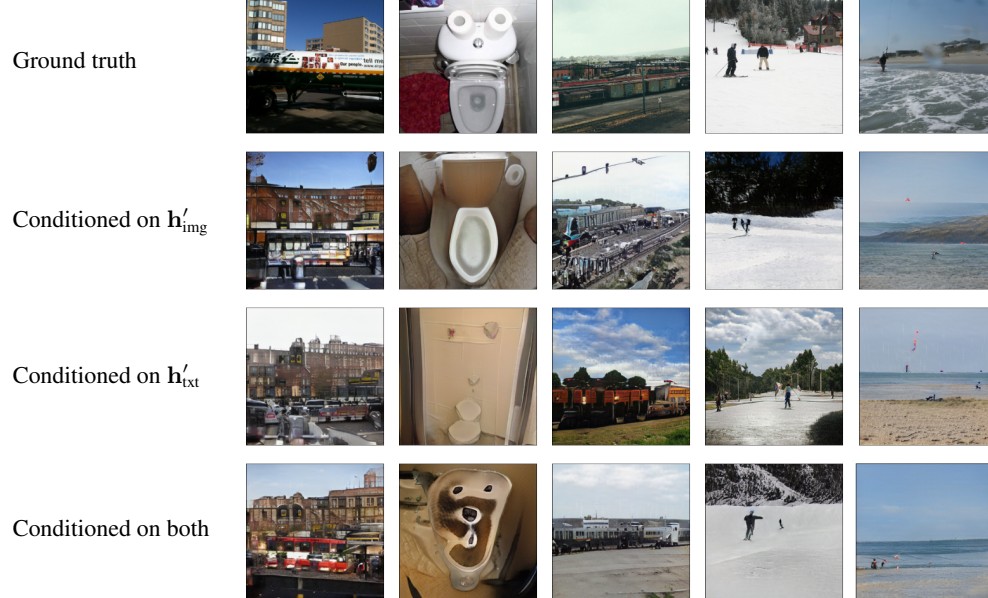

Figure 15: Generated images conditioned on fMRI-mapped CLIP image embedding $\mathbf{h}'_{\text{img}}$, fMRI-mapped CLIP text embedding $\mathbf{h}'_{\text{txt}}$, or both.

The pipeline tends to fail under the following conditions: (1) the image only contains close-up details without too much semantic information; (2) the presented scene is semantically novel (e.g., a big banana-shaped decoration hanging in the middle of the room). The model also tends to: (3) generate based on data biases: adding windows to indoor scenes, adding people to food scenes, generating colored images when the inputs are black-and-white, etc.; (4) change or ignore the background; (5) Mix-up colors (assigning colors in the scene to a wrong object); (6) generate the wrong number of objects/people. We showcase these failures together with more other generated images in fig. 16. Given that the model is confident (in terms of GAN's discriminator output staying at the same level) when generating results based on training data biases, future extensions should focus on exploring generators pre-trained on a much larger dataset, as discussed in A.7.

## A.9 Encoding and encoding-decoding cycle

This paper mainly focused on decoding brain activities. However, we also tested the encoding process with CLIP as the intermediate. In this section, we briefly present our results, as well as the complete encoding-decoding cycle.

**Brain Encoding** Brain encoding is a problem that predicts brain activities from stimuli. It has a data scarcity problem similar to the decoding process. In addition, brain activities are intrinsically noisy and contain randomness, even when responding to the same stimulus. To this end, we solve the problem similar to the decoding process: the image stimuli are passed through pretrained CLIP encoders, obtaining CLIP embeddings $\mathbf{h}_{\text{img}}$. Then we train a mapping model that perform regression from $\mathbf{h}_{\text{img}}$ to $\mathbf{x}_{\text{fmri}}$. The mapping model is also similar to $f_m$, consisting of four residual blocks, one transposed convolutional layer, two linear layers, and is trained with a combination of MSE and cosine similarity loss.

Fig. 17a shows the signal ground truth and predictions for the first 1000 voxels of two samples. We also found that voxel-wise prediction (in terms of the correlation coefficient) aligns very well with the noise ceiling of that voxel (see fig. 17b).[4] However, there are discrepancies in this alignment: in fig. 17c, we visualize the voxel-wise prediction correlation coefficient ($cc$) minus the voxel's noise ceiling ($nc$) as a flatmap. Here, redder areas correspond to better predictions, and the result shows that high-level semantic regions are better predicted than V1-V4. Utilizing latent spaces other than

---

[4]Noise ceiling values are calculated based on the method in the NSD data paper [3], utilizing SNR

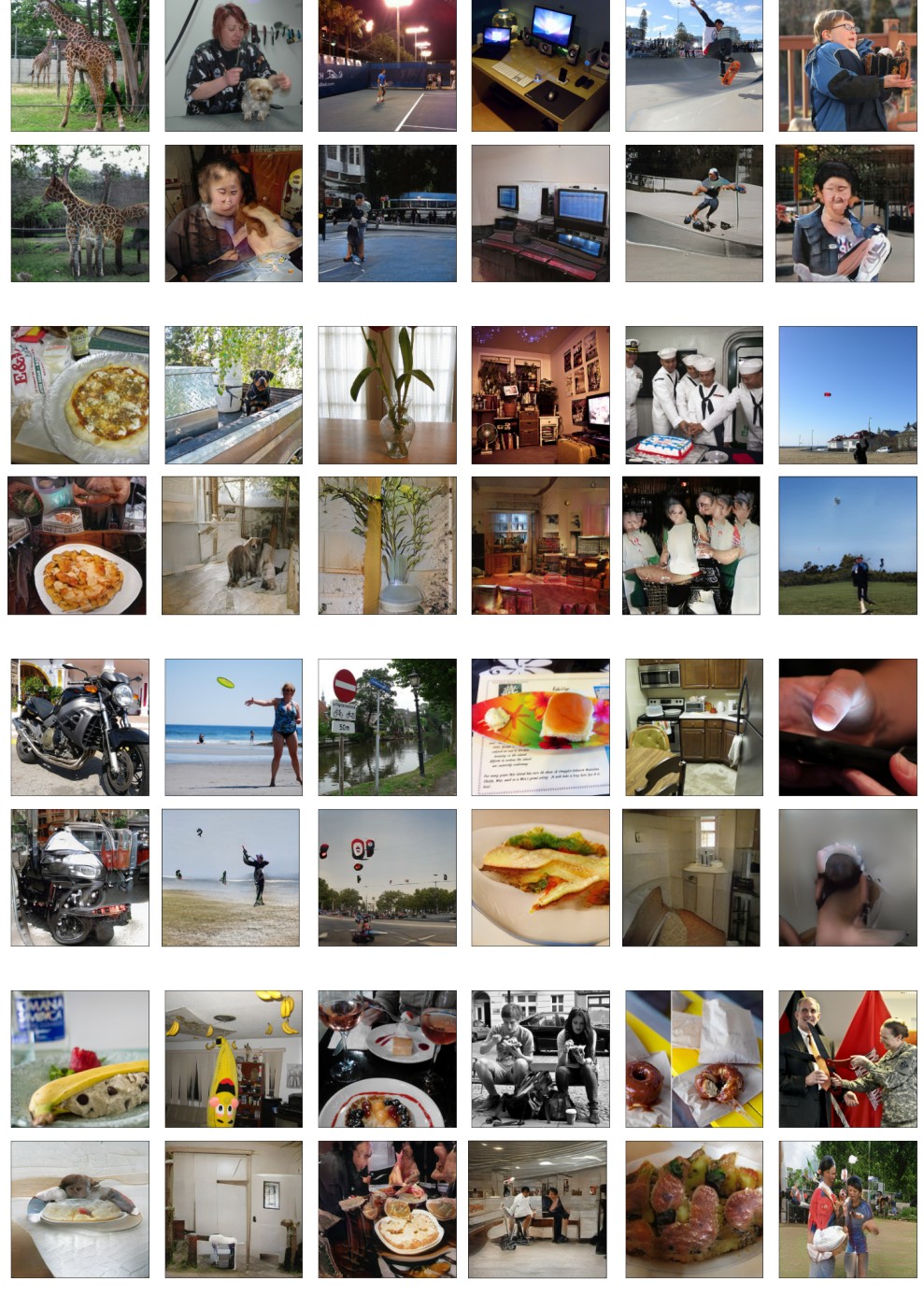

Figure 16: More examples showcasing model successes and failures. For each two-row group, the top row shows the ground truth images, and the bottom row shows the reconstructions.

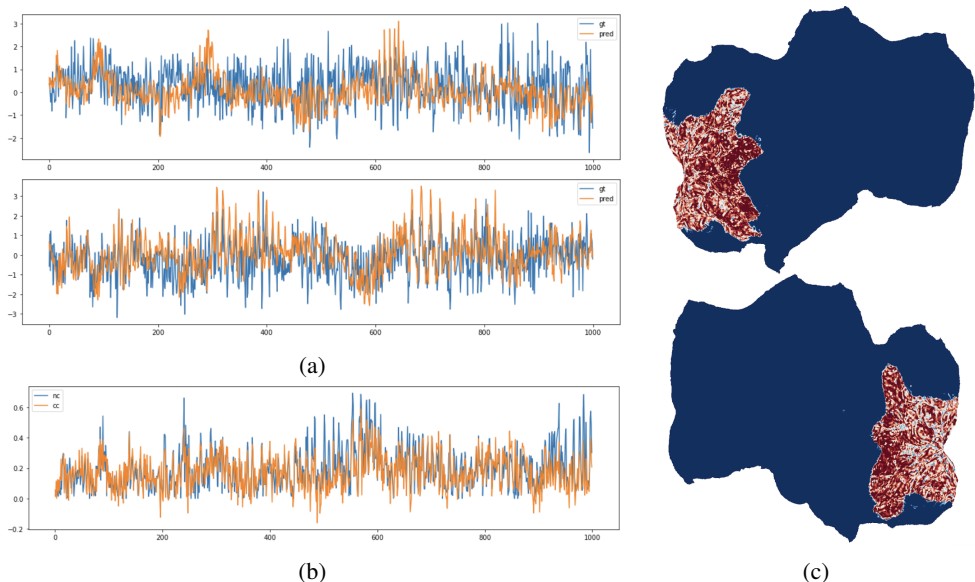

Figure 17: Brain encoding results. 17a ground truth and prediction of two samples. Only the first 1000 voxels are shown for visualization purposes. 17b Voxel-wise performance (in terms of the correlation coefficient between ground truth and prediction) v.s. voxel noise ceiling. 17c Prediction performance on a flatmap, redder regions have more accurate predictions (accounted for the noise ceiling). Note we only perform prediction on the nsdgeneral ROI, thus the boundary.

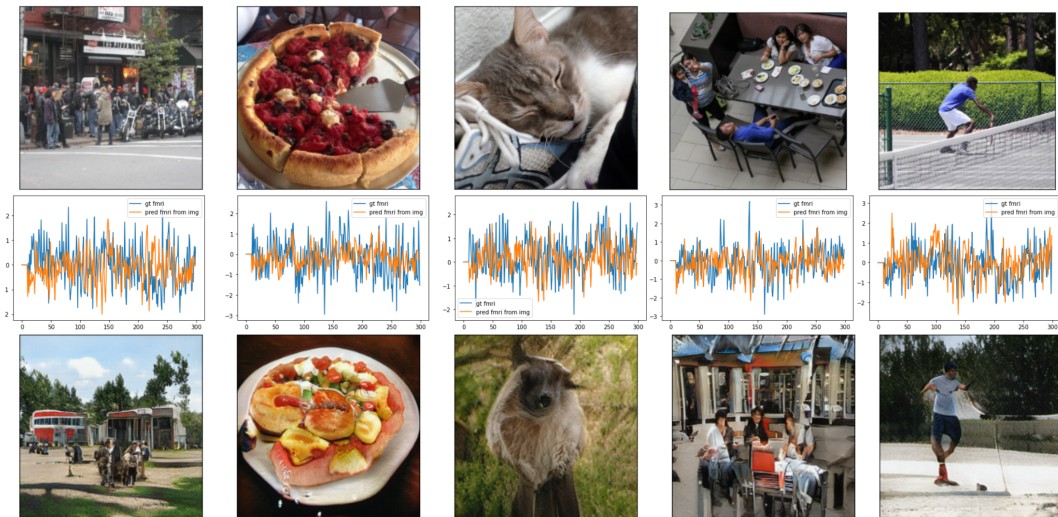

Figure 18: Encoding-decoding cycle. The top row shows image stimuli; the second row shows predicted fMRI activities (with corresponding ground truth) by the encoding pipeline (only 300 voxels are shown for visualization purposes); the third row shows reconstructed images from predicted fMRI signals.

CLIP's results in lower prediction performance and larger distance between $cc$ and $nc$, as well as a more uniform performance among high-level regions and V1-V4.

**Complete Cycle** We tested the encoding-decoding cycle with trained encoding and decoding pipelines: ground truth images are fed to the encoding pipeline, which gives fMRI predictions. We then pass these predicted fMRI signals through the decoding pipeline to perform decoding. The results are shown in fig. 18. We observe that image semantic information is still relatively well conserved.

### A.10 Additional future directions

**Input interpolations and the potential extension to movie reconstruction** In addition to reconstructing observed images, we found utilizing the CLIP space can also result in a smooth transition

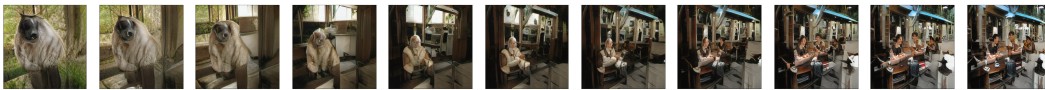

Figure 19: Generated images from interpolation of two fMRI scans. Step number is set to 10.

when decoding from interpolations of two fMRI scans (fig. 19). Combined with the ability to capture complex semantics, this pipeline can be helpful for movie reconstruction from brain signals. Temporal constraints can also be added, which could, in turn, benefit the reconstruction of each frame.

**Decoding text from fMRI** Apart from being the conditional vector for an image generator, CLIP embeddings can also be used to generate texts. To decode texts from fMRI data, the only change needed is replacing the conditional image generator in our pipeline with a text generator conditioned on CLIP vectors.[5] With this text pipeline, one can "define" the functions of each voxel through the following procedures: (1) provide a pseudo-fMRI activity to the pipeline with only the target voxel having non-zero activities, (2) generate fMRI-mapped CLIP embeddings $\mathbf{h}'$ with the mapping models $f_m$, (3) provide $\mathbf{h}'$ to the conditional text generator and get the text description of that voxel activity. An advantage of decoding the signals into the text form is that text is more straightforward than images in terms of explaining the semantics. This makes it easier to perform voxel clusterings and to find brain modules. The texts can also help understand which parts of the semantics are not mapped through from the $f_m$ by comparing the ground truth captions and generated texts from the fMRI activities.

**Neural population control with synthetic images** With the encoding pipeline that we briefed in A.9, one can feed the pipeline with artificial images to test and understand how different shapes and semantics trigger voxels at various locations, thus having a better understanding of voxel functionalities. In addition, works similar to [5] can be tested by finding out which type of stimuli trigger a specific level of brain activity (e.g., higher activation) and then synthesizing images that control the neural population in the desired manner. Lastly, given pipelines of a complete cycle, images generated by the decoder can also be benchmarked by passing them through the encoder.

---

[5]An example CLIP-conditioned text decoder can be found here: https://github.com/fkodom/clip-text-decoder.