# OpenReview forum: "Mind Reader: Reconstructing complex images from brain activities"
_NeurIPS.cc/2022/Conference — NeurIPS 2022 Accept_

### Official Review · Reviewer_WHmz · 2022-06-23

**Rating:** 6
**Confidence:** 4
**Soundness:** 3 good
**Presentation:** 4 excellent
**Contribution:** 3 good

**Summary:**

The authors develop a pipeline to reconstruct natural scene images from fMRI data by aligning neural representations to the CLIP embedding space and then using a generative model conditioned on the CLIP space to reconstruct photo-realistic images.

**Questions:**

By utilizing the pretrained CLIP embedding space as an intermediary, and the pretrained GAN for image generation, brain signals are coerced to adhere to the space of natural statistics learned by these models from their original training sets. This makes it completely unsurprising that photo-realistic images are generated for natural scenes, but leaves the reader wondering what would happen if participants were presented with images that differed from the types of images the pretrained pipeline components have been trained on. How adversarial might these images be? One might expect that the pipeline would continue to generate highly photo-realistic images from whichever part of the training set is most aligned, but these will bear little truth to what the participants actually saw. Do the authors care to comment on this? The image reconstruction examples that the authors choose to show on planes highlight this concern, where spurious details are added to or removed from the reconstructed images, altering the veracity of the images, while assigning unrealistically high confidence to the reconstructions due to their high resolution. In many aspects, on a pixel-level, prior methods are more accurate reconstructions, and don’t create this hyper-realistic false reality. The microstimulation experiments section also highlights this concern, where images are highly degraded from ground-truth in many dimensions that are completely unrelated to the expected region, but still coerced into an overly confident final image. The veracity of resulting images and their proximity to ground truth might be best evaluated using human raters, but I understand that the time involved to crowd-source these ratings is out of scope for the review period.

**Limitations:**

As noted in the previous section, there are considerable concerns to this approach, and the authors neglect discussion of these topics entirely. The authors also fail to address any negative societal implications. By using pretrained models, highly photo-realistic reconstructed images are coerced to fit the natural statistics of the original training sets in ways that are poorly understood, with little regard for the true content of images, and could likely be manipulated to produce offensive or biased images, while attributing those images to the neural signals of the human participant.

Overall, this paper is incredibly creative and well-designed from an engineering perspective, and has incredible potential to be accepted, but needs to be placed into perspective, the limitations section needs to be drastically expanded, and examples should be included of model failure modes, not just best-case scenario reconstructions.

**Strengths And Weaknesses:**

This is a very creative idea, and an interesting approach to generating photo-realistic images from fMRI data despite data scarcity. The proto stimulation and ablation experiment provides particularly fascinating insights into the strengths and weaknesses of this approach, and is both a welcome and exciting addition. In addition, the engineering aspects of this paper are excellent and well-designed. The level of detail is appreciated.

That being said, there are some pretty major fundamental concerns with this approach, in particular the constraints placed on images by utilizing pretrained model components, which the authors entirely neglect to address. Considerable more discussion is required by the authors on this topic, and the associated limitations, and I raise a few specific questions below.

---

> ### Author Response · Authors · 2022-07-30
> **Addressing limitations and the usage of pretrained models**
>
> We really appreciate your comments. Thank you!
>
> You raised a very important concern regarding using the pretrained models. In the revision, we added in-depth discussions in appendix A.8 to address this topic, mentioning the different vulnerabilities of CLIP encoders and conditional GAN. In short, (1) CLIP encoders are trained on a huge dataset (hence very generalizable) and capable of retaining object-level information in the fMRI. However, the length of the embeddings (512) is much shorter than the original signal, thus making it a “compression” of the original signal and more vulnerable to attacks. (2) For the generator, we mentioned the flexibility of their choices: in future works, we can swap them to a generator that is trained on a much larger and unbiased dataset. To illustrate that our system works across different generators but dataset biases are an important issue to consider, we also showed some examples using a CLIP-conditioned generator trained on another dataset (Fig13). All the generated images have the watermark of their training set, but still show good semantic alignments.
>
> We also added a paragraph to section 4 regarding this issue. In it, we discuss the system’s potential negative social impacts in the future, when high spatial-resolution portable brain imaging devices are readily available. We mentioned two tricks to combat the issue: a trained pre-input discriminator to filter the inputs; and a parallel system targeting pixel-level reconstruction (then give a confidence score based on the alignment between our system and this pixel-level system).
>
> In appendix A.9, we showcase some failed results with a short summary about **when** (closed-up details, semantically novel scenes) and **how** (generate based on biases, change/ignore backgrounds, mix-up colors, generate the wrong number of objects) the model tends to give wrong or spurious reconstructions. We hope you find these additions interesting and useful!
>
> In summary, we made the following changes to the manuscript:
> - Extended the discussion on limitations and future work. Section 4 added: DALL-E 2, visual imagery; A.6 added: movie reconstruction. **A.8 added: biases of pre-trained models**.
> - Addressed potential **negative social impacts (Section 4, last paragraph)**.
> - Supplemented (in A.9) with **more reconstruction results** (and images generated with text-only conditions, image-only conditions, and image-text conditions), **including failed cases**.
> - Provide quantitative evidence supporting our main claim: using CLIP embeddings as the intermediary between fMRI and image is beneficial (A.7)).
> - Reorganized the paper: reduced the main text’s reliance on the appendix, making it more self-contained.

---

> > ### Comment · Reviewer_WHmz · 2022-08-08
> > **Additional details and discussion appreciated**
> >
> > Thank you for including a better discussion of failure points, and for expanding the reconstruction results. The new quantitative evidence also improves the soundness of the paper. As such, I have made the following adjustments.
> >
> > Soundness: 2 -> 3
> > Rating: 4 -> 6

---

> > > ### Author Response · Authors · 2022-08-09
> > > **Appreciate the response and increasing scores**
> > >
> > > We greatly appreciate your positive response to the revision and acknowledging our work.

---

### Official Review · Reviewer_as89 · 2022-07-11

**Rating:** 6
**Confidence:** 3
**Soundness:** 3 good
**Presentation:** 3 good
**Contribution:** 3 good

**Summary:**

The authors propose a new method to reconstruct image stimuli from fMRI data. They propose to rely on the CLIP model, and map fMRI signal to the CLIP latent space. Subsequently, they fine-tune a generative model using mapped embeddings so as to reconstruct the seen image.
Experiments show that the approach is able to reconstruct high quality images with high semantic similarity with the target image.
The paper has a several limitations and presentation issues, but overall proposes an original approach that could be of interest to the community.

**Questions:**

-Have you considered the DALL-E 2 model? It combines CLIP with diffusion models and could be a good extension to the proposed approach.

-Have you considered adding a pixel-level loss function to generate images closer to the target image?

-What is the impact of using only the CLIP text or image embedding? Why maxpool values, instead of e.g concatenating?

**Limitations:**

Authors provide a good discussion on limitation. Considering the topic studied in this work, I would recommend to add a discussion on societal impact, and the implications of being able to predict what the brain sees from fMRI data.

**Strengths And Weaknesses:**

**Strengths**

-The idea to rely on large pre-trained models to address the data scarcity inherent to fMRI data is a very interesting idea. The CLIP model is very powerful, and leveraging its high representation power for fMRI is a clever solution.
-The generated images are of high quality, with high semantic similarity with the target image.
-Authors provide an honest and detailed discussion on limitations, including discussion on previous negative results.

**Weaknesses**

-My first issue relates to presentation of the work. Firstly, the paper leverages a lot of diverse concepts and methods (fMRI, CLIP, Lafite model), and assumes a lot of prior knowledge on the reader regarding how these methods work. It would make it difficult for a reader not familiar with the models to understand the methodology.
One key presentation issue is the excessive reliance on supplementary materials. Most of the quantitative results are in the appendix, and some results are directly referred to in the main manuscript. The manuscript in its current form is not self contained. I strongly recommend to rearrange the paper so that this is not an issue anymore.
The paper doesn't include a related work section, which is essential to properly relate the work to the literature.

-The main limitation, which authors are openly discussing, is that the approach does not aim to reconstruct the image, but generate a semantically similar image. While this is interesting and the results on the few examples provided look quite good, this might not be sufficient if certain image details matter (e.g. colours).

-Another issue is the fact that the work is heavily engineered: it comprises a large set of tunable parameters and losses (e.g. 7-8 loss parameters for clip mapping, 6 losses and dedicated tuning parameters for the GAN part), CLIP embedding values are clipped to an arbitrary value, captions are selected based on a threshold, etc..Providing evidence that these parameters do not crucially affect performance would be appreciated.

-The evaluation is mainly qualitative, with limited examples and comparisons to pre-existing approaches. It would have been good to provide additional qualitative examples in the supplementary materials to demonstrate that the shown examples were not cherry picked. For quantitative results, authors propose a new evaluation metric because of a limitation of their approach (ie focus on semantics rather than reproducing the original image).
Several conclusions are made on the basis on a couple qualitative examples (e.g. semantic closeness between mismatches, microstimulation experiments), again, it would be beneficial to provide more evidence to support these claims.

---

> ### Author Response · Authors · 2022-07-30
> **Addressing comments**
>
> Thank you for your very detailed comments! They are all very useful.
>
> To answer your comments:
> - In the revision, we reduced the reliances of the appendix in the main text, hopefully it is now more self-contained.
> - We didn’t include a whole section on related work because of the space limit. However, we briefly discussed them in the introduction and referred readers to the survey [27] when performing the qualitative comparison.
> - We acknowledge the limitation of reconstructing certain low-level features, as written in the discussion section, and would love to study how to incorporate them into the system in the future.
> - It is a valid concern about the number of hyperparameters in the pipeline. We used the sweep feature on wandb (wandb.ai) to tune the 3 loss hyperparameters for mapping models in the first phase. The number is 3 because two modalities share the same set of hyperparameters, and we made all \alpha sum to 1. Since the model isn’t hard to train, and wandb has a sweep option that tunes them according to the bayesian principle, the whole process was rather fast. Wandb also gives the correlation between results and each hyperparameter. For the second phase, we borrowed many hyperparameter values from Lafite, with a few more runs trying different numbers, but we didn’t find they have too much of an impact. Nonetheless, we provided all the values we used in A.2 in case others need them in the future. Regarding clipping the CLIP embedding values, we think this value does play an important role: including it, compared to no-clipping – meaning clipping threshold has the value max(abs(embedding)), increases the performance a lot. But we indeed chose it to be +-1.5 manually by examining the CLIP embeddings and did not experiment with other values.
> - Since we target complex scene reconstruction, it is difficult to compare with pixel-level methods targeting images with single objects. In revision, we provide extra quantitative support in A.7, showing that choosing CLIP embeddings as the intermediary can retain almost all fMRI information regarding the objects in the scene. In contrast, image-only embeddings, like ResNet-50’s, do not. We also provide more generated examples in A.9.
>
> Regarding the questions:
> - About DALL-E 2: Yes! We were very excited about DALL-E 2 when it came out (although its codes/pretrained models weren't available in May). It can definitely be an extension since it also generates images conditioned on CLIP. It also connects the text and image embeddings with a prior model, with which some parts of our current pipeline (choosing captions, clipping embedding values) could be saved. We added it to the revision since its training dataset is much larger, thus having smaller dataset biases than Lafite pre-trained on COCO. The extension to DALL-E 2 would undoubtedly be a promising direction.
> - Regarding pixel-level loss function: Yes, we did try it but failed. It is possible that CLIP space doesn't contain enough information on the pixel level, and additional conditions (that retain pixel information) need to be provided to the generator in order to recover that. However, we haven't found a good embedding space that can retain pixel-level information from fMRI and at the same time have a generator pre-trained on a massive dataset.
> - We mentioned briefly in Section 2.2 that using only one of the embedding modalities tends to emphasize a little more on the generated results’ image or semantic content. The difference is rather subtle, but we provided some examples in A.9 of the revision. As for choosing maxpool, it was just a trial-and-error process: we experimented with average, maxpool, and concatenate + reduce with mlp (the additional reduce is because we need to make the condition tensor back to the previous shape to utilize pretrained weights), and we found maxpool was the best of the three choices.
>
> We also added a paragraph to Section 4 in the revision on its potential negative social impact. In summary, we made the following changes to the revision:
> - **Reorganized the paper: reduced the main text’s reliance on the appendix**, making it more self-contained.
> - **Provided quantitative evidence** supporting our main claim: using CLIP embeddings as the intermediary between fMRI and image is beneficial (**A.7**)).
> - Addressed **potential negative social impacts** (Section 4, last paragraph).
> - Extended the discussion on limitations and **future work. Section 4 added: DALL-E 2**, visual imagery; A.6 added: movie reconstruction; A.8 added: biases of pre-trained models.
> - Supplemented **(in A.9) with more reconstruction results (and images generated with text-only conditions, image-only conditions, image-text conditions)**, including failed cases.

---

### Official Review · Reviewer_X8Bs · 2022-07-11

**Rating:** 6
**Confidence:** 4
**Soundness:** 3 good
**Presentation:** 4 excellent
**Contribution:** 3 good

**Summary:**

This paper proposed a method for reconstructing real-world complex images with both naturalness and fidelity using multimodalities: fMRI, images, and text. The authors encode the fMRI signal into a pre-trained visual language latent space and reconstruct the image using a generative model conditioned on the embeddings. A microstimulation study is performed on different brain ROIs for interpreting human brain functions.

**Questions:**

Please see Limitations.

**Limitations:**

More ablation studies should be listed, for example with/without ROI extraction, and different architectures of mapping model fm.

The authors state 'the generated images emphasize either image content or semantic features depending on which condition we use', more explanation and examples should be done here.

The regularity and variability of using different individual fMRI signals for image reconstruction should be validated and explained.



**Strengths And Weaknesses:**

Strengths:
The evaluation and analysis are relatively comprehensive,  covering different proposed design components and several hyperparameter settings in both training stages.
A good implementation of reconstructing real-world complex images with both naturalness and fidelity from brain activities via fMRI.

Weakness: The models and methods used in this paper are reasonable, but each part of them is known in the field.

---

> ### Author Response · Authors · 2022-07-30
> **Addressing comments**
>
> Thank you for the comments and suggestions!
>
> We acknowledge the individual components of the pipeline are from existing model building blocks, loss designs, or pre-trained models. However, we believe that combining them in a way that actually works, and incorporating additional text modality into brain signal reconstruction is new and very effective, thus providing new insights and opportunities for future studies.
>
> - Regarding the ROI extraction ablation study: we agree that there might be more information that is not included in the “nsdgeneral” mask. But using it is a design choice that makes the model training practical (since the number of voxels would be too large to fit into the memory without a mask) while considering the data quality (during NSD’s experiments, the scanner focused more on the posterior part of the brain, thus voxels under those parts have higher SNR, “nsdgeneral” is a manually drawn mask that includes all voxels that are responsive to the experiments in the posterior aspect of cortex). We did, however, study different ROIs residing in the “nsdgeneral” through microstimulation studies.
> - Regarding the mapping model f_m: we experimented with three types of mapping model, (1) the current one  (Conv1d with residual blocks plus linear layer); (2) a model with two linear layers; (3) a model that maps fMRI latent embedding (obtained from a 1D VQVAE that can reconstruct fMRI signals well) to CLIP with two linear layers. We hoped (3) to work the best since it’s mapping from a shorter input to the CLIP vectors. But in fact, both (1) and (2) outperformed (3), with (1) being slightly better than (2).
> - In the revision, we added more results in A.9 that show images generated conditioned only on (fmri-mapped) image embeddings or text embeddings, or both.
> - By different individual fMRI signals, do you mean fMRI of different subjects? We did mention this in the “further discussions” (section 4) as our current limitation and would love to study differences across subjects.
>
> To summarize the changes we made in the revision, we:
> - **Supplemented (in A.9) with more reconstruction results (and images generated with text-only conditions, image-only conditions, image-text conditions)**, including failed cases.
> - Provided **quantitative evidence** supporting our main claim: using CLIP embeddings as the intermediary between fMRI and image is beneficial (**A.7**)).
> - Addressed potential negative social impacts (Section 4, last paragraph).
> - Extended the discussion on limitations and future work. Section 4 added: DALL-E 2, visual imagery; A.6 added: movie reconstruction; A.8 added: biases of pre-trained models.
> - Reorganized the paper: reduced the main text’s reliance on the appendix, making it more self-contained.

---

### Official Review · Reviewer_cvSx · 2022-07-12

**Rating:** 6
**Confidence:** 5
**Soundness:** 3 good
**Presentation:** 3 good
**Contribution:** 3 good

**Summary:**

In this paper, the authors propose a new method to reconstruct the images from the corresponding fMRI signals by leveraging the text descriptions of the input images and using that as a training signal for the model. The fMRI data for this task is limited and therefore this paper addresses the data scarcity problem by leveraging an aligned vision-language latent space that is pretrained on large-scale datasets. fMRI signals are then aligned to this latent space thus leading to reconstructions that are photo-realistic and capture the semantic contents of the groundtruth image.

**Questions:**

My main concern about this work is its comparison with previous works.
1. It lacks quantitative comparison with any of the proposed framework
2. Qualitative comparison is not fair

**Limitations:**

Limitations and Negative impact section is missing. I have addressed key limitations in the weaknesses section.

**Strengths And Weaknesses:**

### Strength
1. The proposed idea to leverage the latent space of a multimodal model (CLIP) in reconstruction from fMRI signal is new and interesting.
2.  The method is evaluated with clear ablations and relevant metrics thus making the experimental validation of the proposed framework very strong.
3. The paper is well written with a clear description of the method, caption selection and different components of the model used.

### Weakness
1. The comparison to previous work is not fair. Previous methods were trained using a smaller different dataset while the model in the present work was trained using a higher SNR dataset. Therefore, whether the proposed method is better than any of the previous works can not be inferred based on this work and only on qualitative results in Figure 5
2. The photorealism comes from the pretrained generator and the previous work[22] which uses BigBiGAN's latent space can also generate photorealistic results. Therefore, a key claim that it is possible to generate photorealistic images using the proposed framework is misleading.
3. There is no quantitative comparison with any of the previous methods. At least one or two strong baselines are easy to implement and should be compared with e.g. [22] or an end-to-end GAN[32] to figure out how competitive is the proposed approach against these baselines.
4. Microstimulation results are not convincing.The behavior of images generated is interesting but there is no quantifiable result to interpret what exactly is happening there.

---

> ### Author Response · Authors · 2022-07-30
> **Quantitative support added to revision A.7**
>
> Thank you for the comments!
>
> We agree the comparisons with previous works are mainly qualitative, but the main reason is that our work targets a new area of reconstruction (complex scenes), making a direct comparison difficult.
>
> Previous works either (1) focus on pixel-level reconstruction, which is totally valid if the goal is to reconstruct a single object in the scene, or (2) use a latent space of image-only modality.
> - Contrary to (1), our work focuses on semantic reconstruction since complex scenes contain more information. This makes both pixel-level metrics and semantic-level metrics unfair. It is also not fair to apply previous pixel-level reconstruction methods to these complex scenes even if the SNR of NSD is higher as its images are also more complex. We did, however, present N-way identification results in the appendix as the quantitive result, which previous works also report.
> - Photo-realistic is just one aspect that we target when making design choices. We agree that when pretrained GANs are used as the decoder, the generated results all tend to be photo-realistic. In fact, the generators of these methods are quite flexible and interchangeable (or even can be swapped for better generators in the future, as we mentioned in the discussion section); it is the choice of latent space that serves as the bottleneck in all these situations. Ideally, we can (a) use the embedding in this latent space to tell if certain objects are observed, and (b) lose as little information as possible when translating fMRI into this latent embedding. The previous methods utilizing GANs [22][32] use image-modality-only space: for example, the encoder of BigBiGAN is ResNet-50, and [32] uses VGG19.  **Our main claim is that implicitly adding text modality into the equation by utilizing the CLIP space is more beneficial than using an image-only space** since it’s more semantically expressive. In the revision appendix A.7, we show, quantitatively, that CLIP space contains more object-level information than fMRI, AND  can retain about 98% of fMRI information when the brain signals are mapped into it. In contrast, ResNet by itself already contains less information than fMRI. This suggests that utilizing BigBiGAN's space, no matter how good the generator is, will lose more information than using the CLIP space. Therefore, we believe our proposal,  utilizing CLIP space, is a very useful step forward. (p.s. We didn't include [22]’s reconstruction in figure 5 because it doesn't show a plane in their reconstruction (fig 3 in their paper https://arxiv.org/pdf/2001.11761.pdf).)
>
> For the microstimulation, since COCO doesn't have "face," "body," etc., categories, and each image tends to have multiple of them, we found it impossible to separate samples into individual task groups. In addition, off-the-shelf detectors, if trained on COCO, won't detect objects as "faces" or "words"… These together make quantifying the stimulation results much more tricky. Therefore, we decided to show the visual results and discuss our findings.
>
> We did mention our limitations in “further discussions” (section 4): (1) our current pipeline focuses more on the semantic alignment in the reconstruction than on pixel-level details, so scenes that contain finer details (like face features) are harder to reconstruct. We believe that this is a valid trade-off for now since subjects need to grasp the essence of the presented image in a short amount of time, and focusing on semantics is a more natural way to do so. However, future works are definitely needed to incorporate more pixel-level details into the reconstruction. (2) our current work models one subject; future work is needed to incorporate more subjects’ data (map signals into a common space, train a shared fMRI to CLIP embedding mapper, etc.).
>
> In the revision, we added more limitations and future work discussions and addressed the negative impact. In summary, we made the following changes to the revision:
> - Provided **quantitative evidence** supporting our main claim: using CLIP embeddings as the intermediary between fMRI and image is beneficial (**A.7**)).
> - Addressed **potential negative social impacts** (Section 4, last paragraph).
> - **Extended the discussion on limitations** and future work. Section 4 added: DALL-E 2, visual imagery; A.6 added: movie reconstruction; A.8 added: biases of pre-trained models.
> - Reorganized the paper: reduced the main text’s reliance on the appendix, making it more self-contained.
> - Supplemented (in A.9) with more reconstruction results (and images generated with text-only conditions, image-only conditions, image-text conditions), including failed cases.

---

> > ### Comment · Reviewer_cvSx · 2022-08-03
> > **Appreciate adding quantitative analysis, increasing score**
> >
> > Thank you for providing clarifications and quantitative analysis showing CLIP embeddings as the better choice for mapping fMRI responses. The response has convinced me to improve my score as I think this paper shows new insights that can help in guiding new works focusing on reconstructing images from neural (fMRI) responses.
> >
> > I have further doubts and would appreciate it if authors provide clarifications
> >
> > ```This makes both pixel-level metrics and semantic-level metrics unfair.```
> > I agree, but the complimentary success of both types of approaches in corresponding metrics can emphasize how combining these two losses (pixel and semantic) in some way may lead to a more accurate reconstruction that is accurate both semantically and in the positions of the objects.
> >
> > ```It is also not fair to apply previous pixel-level reconstruction methods to these complex scenes even if the SNR of NSD is higher as its images are also more complex```
> > Why not? I agree that they won't work as well as the proposed approach, and therefore, it would provide another evidence supporting the semantic reconstruction.

---

> > > ### Author Response · Authors · 2022-08-04
> > > **Appreciate the additional comments and increasing the score**
> > >
> > > Thank you for acknowledging our work!
> > >
> > > We agree that both types of metrics could provide extra insights into the reconstruction performance, and comparing with pixel-level methods can show the strength of our pipeline for semantic reconstructions. Since our work targets more complex scenes, the comparison should be carried out on the NSD dataset. Therefore, we cannot directly use the numbers provided in the previous papers. We will add results from [5], [32] (they are the better-performing methods under each metric based on the survey [27]), and potentially [22].  We do need more time to revise and train them on NSD as their codes are not up-to-date with current ML libraries. But the comparison should be in our final version. Thank you again for your time and comments!

---

### Meta-Review · Area_Chair_8KSb · 2022-08-27

**Recommendation:** Accept
**Confidence:** Certain

**Metareview:**

In this paper, the authors propose a new method to reconstruct the images from fMRI signals by leveraging the text descriptions of the input images and using that as a training signal for the model. The fMRI data for this task is limited and therefore this paper addresses the data scarcity problem by leveraging an aligned vision-language latent space that is pretrained on large-scale datasets. fMRI signals are then aligned to this latent space, leading to reconstructions that are photo-realistic and capture the semantic content of the ground truth image.

The reviewers agreed that the proposed idea to leverage the latent space of a multimodal model (CLIP) in reconstruction from fMRI signal is new and interesting. At the same time, they voiced concerns about some of the evaluation metrics and baselines. Over the course of the rebuttal, many of the reviewer concerns were addressed (at least mostly) and reviewers increased their scores. In the end, all of the reviewers agreed in favor of acceptance.


**Award:**

No

---

### Decision · Program_Chairs · 2022-09-14

Accept